# Intrinsic neuronal dynamics predict distinct functional roles during working memory

D.F. Wasmuht[1,2], E. Spaak[1,2], T.J. Buschman[3], E.K. Miller [4] & M.G. Stokes[1,2]

Working memory (WM) is characterized by the ability to maintain stable representations over time; however, neural activity associated with WM maintenance can be highly dynamic. We explore whether complex population coding dynamics during WM relate to the intrinsic temporal properties of single neurons in lateral prefrontal cortex (lPFC), the frontal eye fields (FEF), and lateral intraparietal cortex (LIP) of two monkeys (*Macaca mulatta*). We find that cells with short timescales carry memory information relatively early during memory encoding in lPFC; whereas long-timescale cells play a greater role later during processing, dominating coding in the delay period. We also observe a link between functional connectivity at rest and the intrinsic timescale in FEF and LIP. Our results indicate that individual differences in the temporal processing capacity predict complex neuronal dynamics during WM, ranging from rapid dynamic encoding of stimuli to slower, but stable, maintenance of mnemonic information.

[1] Department of Experimental Psychology, University of Oxford, Oxford OX1 3UD, UK. [2] Oxford Centre for Human Brain Activity, Wellcome Centre for Integrative Neuroimaging, Department of Psychiatry, University of Oxford, Oxford OX3 7JX, UK. [3] Princeton Neuroscience Institute and Department of Psychology, Princeton, NJ 08544, USA. [4] The Picower Institute for Learning and Memory and Department of Brain and Cognitive Sciences, Massachusetts Institute of Technology, Cambridge, MA 02139, USA. Correspondence and requests for materials should be addressed to M.G.S. (email: mark.stokes@psy.ox.ac.uk)

Single-neuron dynamics in higher cortical areas are heterogeneous, complicating interpretation of their functional roles during cognitive tasks[1,2]. A prominent example illustrating this principle is working memory (WM), a mental process strongly associated with the prefrontal cortex (PFC)[3–5]. During WM, information about a transiently encoded stimulus needs to be stored and kept available over a short delay before a response can be made[6,7]. The neural mechanisms by which this kind of cognitive stability is achieved remain a matter of scientific debate[8–10].

A large body of experimental studies emphasize a coding scheme whereby information is maintained through persistent firing of single neurons[3,11–13]. This view is supported by theoretical models demonstrating that persistent activity can emerge from either intrinsic cell properties[14,15] or reverberations in recurrently connected populations of selective neurons[16–20].

However, other studies highlight heterogeneous temporal tuning profiles in a majority of recorded neurons[21–23] or distinct bursts of activity[24], resulting in a highly dynamic population code underlying WM[25–27]. Supported by theoretical models[28–30], those observations have led to the view that maintenance of stable mental representations in WM is possible in the absence of persistent activity in single cells, either through coordinated transient dynamics[30] or rapid connectivity changes[29,31].

Given the frequent observations of both persistent and dynamic coding at both the population and single-neuron level, it is unlikely that these ideas are mutually exclusive[8,32]. In fact, recent studies have shown that stable population coding can coexist with heterogeneous neuronal dynamics[27,33]. While those studies stress stable population coding despite overall heterogeneous neuronal dynamics, critically, both regimes interact during categorizations performed by monkeys during WM tasks, relying on both persistent and dynamic neurons[34]. Both transient and persistent activity in single neurons seems to be important for WM. However, the source and function of the neuronal tuning and temporal variability underlying WM population dynamics remain poorly understood.

One of the most striking principles observed across the mammalian cortex is the hierarchical organization of the receptive field size. While this phenomenon has been well studied in the visual system, where spatial receptive field size increases with functional hierarchy[35], a similar principle is currently being uncovered for the temporal domain[36]. Murray et al.[37] estimated decay time constants for neurons' spiking autocorrelation during baseline activity, coined "intrinsic timescale", across areas in monkey cortex. They reported that intrinsic timescales increased along the cortical hierarchy. This observation was further supported by evidence from previous studies in monkeys and mice[38,39] and results from human imaging studies[40,41]. Importantly, a recent modeling study demonstrated that the observed gradient of intrinsic timescales arises naturally in a large-scale brain model through the balance of intra- and inter-area connection densities[42].

Intrinsic timescales can be interpreted as the duration over which cells integrate information[43]. According to this view, shorter timescales in sensory areas promote rapid detection of dynamic stimuli[44,45], increasing the temporal dimensionality of neural coding. Conversely, longer timescales in prefrontal areas could support integration of information over longer periods of time and could improve signal-to-noise ratio needed for decision-making or WM[43,46], at the cost of dimensionality in the temporal domain. Consequently, heterogeneity in intrinsic timescales found within a cortical area might reflect functional specializations of individual neurons according to temporal processing demands. In line with this, a recent study by Scott et al.[47] showed that the fronto-parietal network encodes accumulated evidence with a diversity of neuronal timescales. A similar observation was made by Bernacchia et al.[48], who found a reservoir of timescales for reward integration across cortical neurons. Furthermore, orbitofrontal cortex (OFC) neurons associated with a relatively long intrinsic timescale were shown to carry the chosen value signals over longer time periods than their short-timescale counterparts[49]. Similarly, Nishida et al.[50] showed that a more stable baseline activity was associated with higher firing rates during the delay period of a WM in lateral intraparietal cortex (LIP).

Here, we explore how the temporal stability of individual neurons at rest (i.e., pre-trial fixation period) relates to the heterogeneity in population dynamics observed during the subsequent WM trial. We tested whether a neuron's intrinsic timescale determines its functional role during WM in three brain regions: the lateral PFC (lPFC), the frontal eye fields (FEF), and the LIP. We find that neurons associated with a relatively long intrinsic timescale carry more information about task-relevant features than short-timescale neurons. Importantly, in prefrontal areas, long-timescale cells also carry information in a more stable way. This is observed for each task epoch, and is especially striking during the delay period, for both individual cells and at the population level. In addition, lPFC cells with shorter intrinsic timescales signal item information earlier during memory encoding and show richer dynamics during the delay period, suggesting specific functions along distinct neural dimensions. Last, functional connectivity at rest is correlated with intrinsic timescales in FEF and LIP but not in lPFC, potentially implicating different mechanisms determining a neuron's temporal processing characteristics.

## Results

**Task and recorded sample.** Spikes were recorded from the lateral prefrontal cortex (lPFC, $n = 583$), frontal eye fields (FEF, $n = 323$), and lateral intraparietal cortex (LIP, $n = 281$) (Fig. 1a, brain schematic), while monkeys performed a delayed change detection WM task (Fig. 1a, left panel). To initiate a trial, monkeys had to fixate a red circle in the middle of a black screen (fixation period: 500 ms). Upon successful fixation, monkeys were presented with a sample array for 800 ms (sample period). A sample array could contain two to five colored squares (items) distributed over six locations (three in each visual hemifield). After a memory delay of 800–1000 ms (delay period) a test array appeared on the screen. The test array was equal to the sample array except for one of the items (the target), which changed color. Monkeys had to indicate the location of the target with a saccade. For details regarding the task structure, refer to the Methods section and to Buschman et al.[51].

**Defining neural populations by their intrinsic timescale.** Our main question was how the intrinsic temporal stability of a neuron[37] contributes to the dynamic processes underlying a WM task. To investigate this question, we computed the decay time constant of each neuron's autocorrelation function during the fixation period[37,42,49]. The resulting quantity is referred to as the intrinsic timescale (tau, τ). A relatively long τ indicates stable firing during fixation, as opposed to a short τ indicating more dynamic baseline-firing patterns. Figure 1b shows two example cells from lPFC with different τ's. We plotted each cell's τ as a function of brain region (Fig. 1c). Average τ values differed across brain regions (Kruskal–Wallis, $p < 0.0001$). Intrinsic timescales in lPFC were larger than in LIP (Kruskal–Wallis followed by Dunn's test, $p < 0.001$), but not FEF ($p = 0.08$). Average τ values in LIP and FEF were not significantly different from each other ($p = 0.14$). Importantly, we also found that our observed τ values did

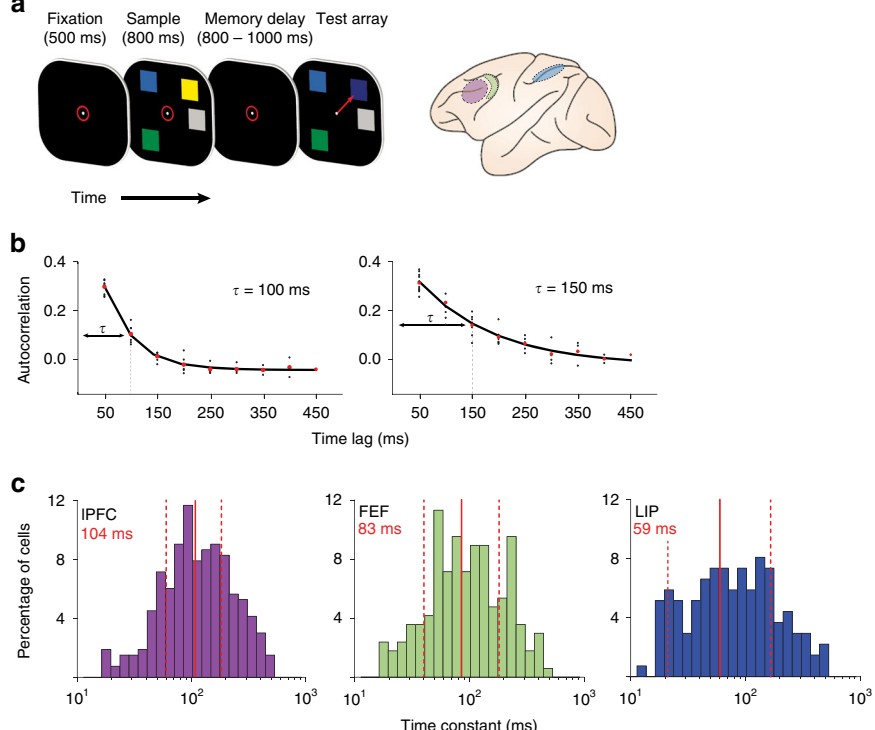

**Fig. 1** Experimental paradigm and estimation of intrinsic timescales. **a** Delayed change detection task: Subjects fixated to initiate the start of a trial (red circle, 500 ms). A sample array was presented for 800 ms consisting of two to five items. After a memory delay (800–1000 ms) a test array was displayed that was identical to the sample, except one item (the target) had changed color. Animals indicated target location by a saccade (see ref.[51]). The right panel shows recording sites: lPFC (purple); FEF (green); and LIP (blue). **b** Autocorrelation decay functions of two example cells estimated over the duration of the fixation period (left: short τ; right: long τ). The red points denote the mean autocorrelation for a specific time lag. Black dots mark the autocorrelation of each time-bin t and any other time-bin t′ at the lag specified on the x-axis. Solid black lines denote exponential fits to the red points. **c** Histograms of the distribution of intrinsic timescales estimated separately for each cell and each brain region lPFC (n = 265); FEF (n = 168); LIP (n=136). Color code corresponds to the brain schematic. Red vertical lines denote mean and standard deviations, estimated from log- transformed τ values (mean values are displayed in red)

not depend on the average fixation firing rate in any of the brain regions (Spearman rank correlation of τ with firing rate; lPFC: $r = 0.06$, $p = 0.3$; FEF: $r = 0.12$, $p = 0.1$; LIP: $r = 0.08$, $p = 0.3$), nor was there any significant relationship between tau and firing rate within each task epoch (Supplementary Figure 1).

**Cells with longer intrinsic timescales carry more information.** To characterize the influence of individual intrinsic firing stability on task involvement, we first quantified the amount of information each cell carried about the task-relevant features (i.e., memory item). Here, we used the percentage of variance-explained (ωPEV) statistic to measure the extent to which the variability in neural firing rate was determined by color and location[51].

To determine the relationship between intrinsic timescale and single-cell item selectivity, we sorted cells by τ and plotted their respective ωPEV over time (Fig. 2a). Visual inspection of the sorted neural populations reveals a clear pattern toward stronger item encoding as a function of increasing τ in all three brain regions. Splitting the neural population according to their respective median τ value and averaging the ωPEV for each median split draws out this conclusion (Fig. 2b). In lPFC, long τ cells carried more information, especially during the delay period (cluster-based permutation test, $p = 0.003$), while in FEF, long τ cells were more informative throughout both the sample and delay periods ($p = 0.008$; $p = 0.02$). Interestingly, there was little difference between cell types during the initial transient responses associated with memory encoding, or the response probe at the

end of the trial: the functional benefit for long τ cells only emerges during the more stable task epoch. LIP cells showed a similar trend during the sample period, but not during the delay period, where mnemonic coding was poor.

To further investigate the observed relationship, we computed a rank correlation between each cell's τ and ωPEV value for each time point in the trial (Fig. 2c). The positive correlation time courses confirm our results. In lPFC, the correlation between τ and ωPEV shows an initial, transient bump (cluster-based permutation test, $p < 0.01$), as also seen in the median-split plots (Fig. 2b). This is followed by a progressive increase during the delay period ($p < 0.001$). In FEF, we observed fast-rising correlation values during the sample period, which stayed high throughout the task ($p = 0.002$). In LIP, the correlation between ωPEV and τ is less straightforward, showing a trend for larger correlation values during the sample period of the task (no cluster survived thresholding), followed by a drop-off during the delay period. Although not directly correlated with τ, the average firing rate during fixation might still influence the relationship between τ and the ωPEV. Therefore, we regressed the average ωPEV estimated from the periods of significant Pearson correlation (Fig. 2c) against τ factoring out the average fixation firing rate (lPFC: $\beta = 0.19$, $p = 0.004$; FEF: $\beta = 0.25$, $p = 0.008$; LIP: $\beta = 0.2$, $p = 0.017$).

Furthermore, we assessed the proportion of cells with significant item information within our tau splits. Within each region, a larger proportion of long tau cells showed significant ωPEV at any time point in the trial: lPFC (long tau: 113/132;

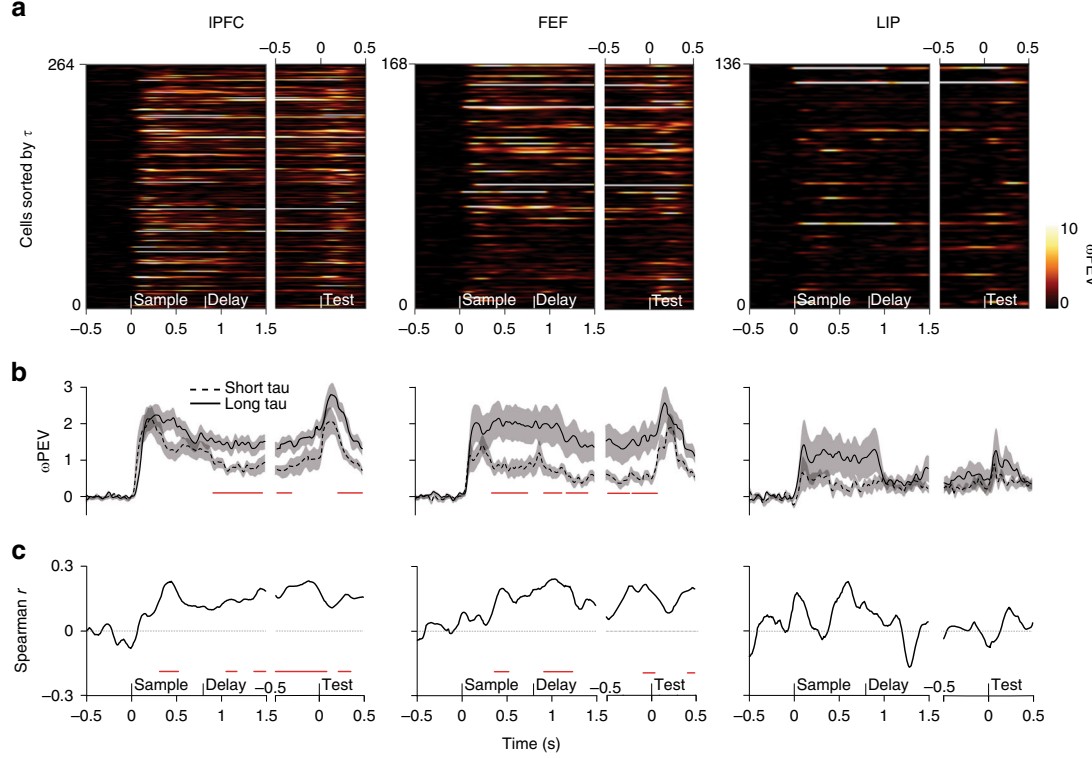

**Fig. 2** Item information and intrinsic timescales. **a**, **b**, **c**: **a** Cells sorted by their intrinsic timescale. The color scale indicates the amount of item information, i.e., ωPEV present at each time point in a given cell. Traces are smoothed with a Gaussian kernel (s.d. = 50 ms) along the time axis and nearest-neighbor interpolation along the cell axis. Initially, trial time is shown with respect to sample onset. However, the duration of the memory delay is variable (800–1000), therefore, we show the last part of the trial (late memory delay until the execution of behavioral response) separately, with trial time shown with reference to presentation of the test screen. The same convention is used for all brain areas (lPFC: n = 265; FEF: n = 168; LIP: n = 136). **b** Median splits of cells according to their intrinsic timescales. The black dashed line denotes the time-resolved average over all cells belonging to the first half of the split, i.e., short τ. The black solid line denotes the time-resolved average ωPEV over all cells belonging to the second half of the split, i.e., long τ. The shaded area represents the standard error of the mean (s.e.m.). The red solid line marks a significant difference between the two splits. **c** Spearman rank correlation of the intrinsic timescales and the time-resolved ωPEV. The red solid line marks a significant correlation coefficient. Correlation time courses were smoothed with a Gaussian kernel (s.d. = 50 ms); statistics were computed on unsmoothed traces

short tau: 90/132), FEF (long tau: 63/84; short tau: 58/84), and LIP (long tau: 29/67; short tau: 24/67). When focusing on the significant cells only, while performing a new median split by tau within those, average ωPEV traces resembled the ones computed from the original cell population (Supplementary Figure 2).

**Intrinsic timescale predicts encoding onset in lPFC**. Cells associated with longer τ values carried more information, which was especially prominent during the delay period in lPFC. In the visual system, smaller receptive fields are useful for detection of rapid fluctuations in stimuli[44]. Analogously, we considered whether short τ cells might therefore be faster at encoding at the time around stimulus onset, since such a division would point toward a differential weighting of cells according to τ for either fast perceptual encoding or longer-term storage of the stimulus.

To investigate this hypothesis, we limited our analysis to cells that showed significant item encoding (ωPEV) at any point during the sample period, since this allowed us to quantify significant encoding onsets. Visual inspection of the average ωPEV in short and long τ subpopulations suggests that short τ cells might encode more information earlier during the sample period in lPFC (Fig. 3a; no cluster survived thresholding). To examine this in more detail, we computed the fraction of currently significant cells for each time point, while averaging over τ values for this fraction (Fig. 3b). We found that, especially during the early period of the sample epoch, as more cells became

task-selective, the average τ value for those task-engaged cells increased as well. This observation was complemented by a significant positive correlation between τ and encoding onset time (time of the first significant ωPEV) during the time period in which the fraction of encoding cells reached 80% of its maximum (first 200 ms of the sample period) (Spearman rank correlation, r = 0.36, p < 0.001). This correlation remained significant even after taking the whole sample period into account (r = 0.16; p = 0.03). This result clearly shows that cells with a shorter τ actively encode task information earlier during stimulus presentation in lPFC. In FEF and LIP, we found no such relationship (Supplementary Figure 3).

**A more stable code for cells with longer intrinsic timescales**. Next, we extended our analysis to investigate how τ influences coding dynamics on the level of the neuronal population. Recent studies successfully used cross-temporal pattern analysis to characterize dynamics during a range of WM tasks[25–27,33]. Here, we applied the same method to investigate potential differences in our two subpopulations of neurons (short vs long τ). In brief, we split the observed trials into two independent halves and computed the average firing rate, per condition, per neuron, for each half. We then computed all pairwise differences within each split, and correlated the pattern of pairwise condition differences of the whole population at each time point **t** in split one with the pattern at every time point **t′** in split two, yielding a two-dimensional

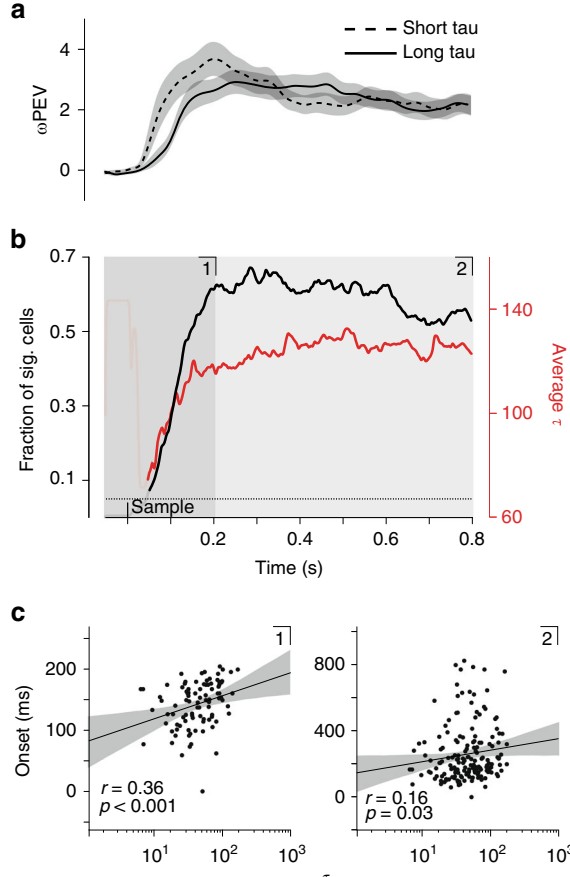

**Fig. 3** Evaluation of sample period onset times in relation to intrinsic timescales for lPFC. **a** Average ωPEV per median split (dashed: short τ; solid: long τ) for sample period only; the shaded region indicates s.e.m. **b** Black line indicates the fraction of significant cells (ωPEV; cluster-based permutation test, $p < 0.05$) at each time point from all cells showing a significant ωPEV at any point during the sample period. The red line shows the average τ value of cells having a significant ωPEV at that time point. The shaded region marks the time period at which the fraction of significant cells first reached 80% of its maximum value. Occluded parts mark the first time period with fewer than ten significant cells. Numbers in the upper part of the panels refer to **c**. **c** Left panel: Scatter plots show onset times (ms) of significant ωPEV estimated during time period **1**, i.e., until the fraction of significant cells reached 80% of its maximum value, versus log-transformed τ values (Spearman rank correlation, $r = 0.36$, $p < 0.001$, $n = 96$). Each dot marks a cell. Right panel: The same scatter plot but with onset times estimated from the entire sample period **2** (Spearman rank correlation, $r = 0.16$, $p = 0.03$, $n = 176$). Black lines indicate linear fits to the data with the shaded area depicting the 95% confidence interval of the fit

matrix representing a discriminability score (see Methods). If the discriminating pattern is stationary over time, the within-time correlation values should resemble the between-time-point ones; i.e., the pattern should cross-generalize over time. Conversely, a dynamic pattern will result in between timepoint correlation values lower than the those for corresponding time points along the diagonal of the matrix.

To probe whether our previous univariate observations generalize to the population, we first applied the above-described decoding approach within time, i.e., for pairs of the same time points between splits (Fig. 4a). Across regions, within-time discriminability time courses strongly resembled those from the ωPEV analysis (Fig. 4a). In lPFC, short τ cells initially showed a higher discriminability score (cluster-based permutation test, $p$

< 0.01) which decayed during the delay period, whereas mnemonic information was more strongly represented by long τ cells ($p < 0.01$). In FEF, long τ cells showed higher discriminability throughout the task ($p < 0.01$), while in LIP, item information was only discriminable during the sample period. Here, again, long τ cells showed higher item discriminability than short τ cells ($p < 0.01$).

Figure 4b shows the across-time extension of our decoding approach. Note, the diagonals of the respective matrix plots correspond to time courses in Fig. 4a. In general, cross-temporal population coding was more prominent and temporally stable for the long τ subpopulation. In lPFC, white contours indicating a significant discriminability score (cluster-based permutation test, $p < 0.001$) extend over the entire task duration for the long τ subpopulation (Fig. 4b). In contrast, in the short τ subpopulation, off-diagonal discriminability drops to chance level at the transition between sample and delay periods and also within the delay period. Subtracting the discriminability matrices computed for the two subpopulations (long τ–short τ) reveals significantly stronger and more stable decoding during the delay period within long τ cells (cluster-based permutation test, $p < 0.01$). In FEF, the discriminative pattern of the long τ cells strongly cross-generalizes both within task epoch as well as across task epochs ($p < 0.001$). This strong cross-generalization is absent in the short τ subpopulation which shows significantly less cross-temporal decoding ($p < 0.01$). In LIP, differences between τ populations are confined to the sample period where long τ cells show a stronger discrimination score ($p < 0.01$), which is remarkably stable ($p < 0.001$).

To illustrate population decoding for both item features separately, we conducted within- and across-time temporal discriminability analysis for location and color only (see Methods) (Supplementary Figures 4 and 5; for statistical test values refer to figure legends). For locations only, the differences between decoding in τ subpopulations in lPFC and FEF were similar to those observed for combined item decoding (i.e., color and location; Fig. 4) (Supplementary Figure 4). In LIP, location information was weak but present during the delay period (cluster-based permutation test, $p < 0.05$), as evident from the cross-temporal plots. Interestingly, long τ cells showed a similar re-emergent pattern during sample and late delay periods. Apart from a very weak effect in FEF, color information was only present in lPFC (Supplementary Figure 5). Here, the long τ subpopulation showed stronger within- and across-time discriminability, especially during the late sample and early delay periods. Color encoding was only weak in FEF and absent in LIP, preventing an informative comparison between τ subpopulations.

Last, we tested whether the decoded information is a behaviorally relevant readout. To this end, we implemented a correct versus error trial analysis. More specifically, we wanted to know whether there was any difference between the decodability of the item that was going to change its color, i.e., the target, on trials where the monkey indicated the correct location of the color change (correct trial) and when it indicated the wrong location (error trial). Our correct versus error trial analysis revealed a significant decrease in target decoding during the delay period in lPFC on error trials (Supplementary Figure 6, cluster-based permutation test, $p = 0.035$). This effect was also clearly visible on the off-diagonals of the cross-temporal plots, where target information of the sample period was similar to that in the delay period only within correct trials (Supplementary Figure 6). There was a similar trend in FEF (Supplementary Figure 6) but our data was too noisy to make any interpretations for LIP (Supplementary Figure 6). However, this observation confirms the behavioral relevance of neural activity in lPFC captured by our decoding approach.

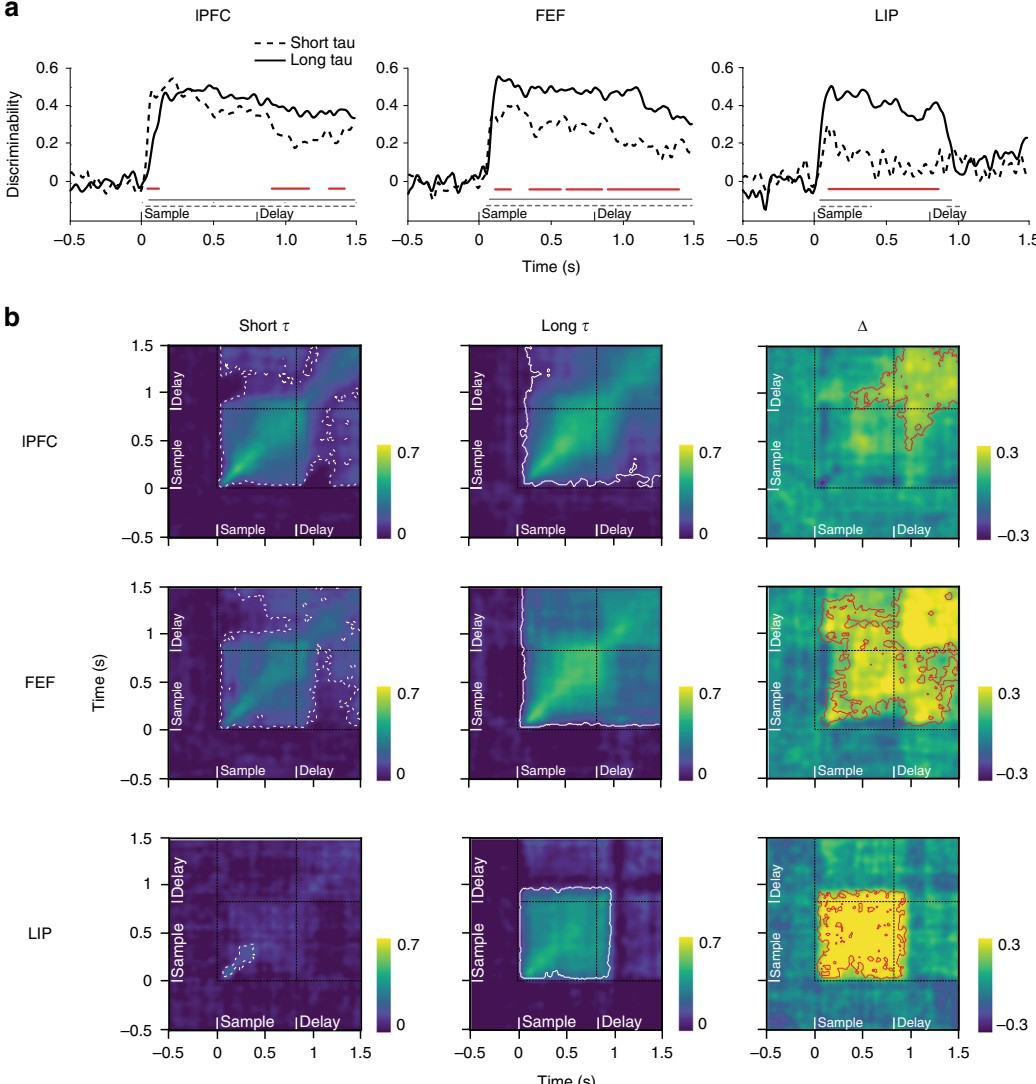

**Fig. 4** Within-time and cross-temporal decoding and intrinsic timescales. **a** Within-time decoding. Solid black lines indicate decoding discriminability for long $\tau$ cells. Dashed black lines indicate decoding discriminability for short $\tau$ cells. Gray lines indicate significant discriminability (cluster-based permutation test, $p < 0.001$). Red lines indicate significant differences between the short and long $\tau$ cells (cluster-based permutation test, $p < 0.01$). **b** The cross-temporal discriminability is color coded. Yellow colors indicate strong decoding, while blue colors indicate weak or no decoding. White dashed and white solid contours indicate significant cross-temporal discriminability within the respective subpopulation (cluster-based permutation test, $p < 0.001$). Black dashed lines distinguish task periods. First column: short $\tau$ subpopulations (dashed). Middle column: long $\tau$ (solid). The right column shows the difference between the two first columns, i.e., long $\tau$–short $\tau$. Red contours mark a significant difference (cluster-based permutation test, $p < 0.01$): lPFC ($n = 265$); FEF ($n = 168$); LIP ($n = 136$)

**Intrinsic timescales predict temporal coding dimensionality.** So far, our results show the existence of a link between a cell's intrinsic firing stability and the magnitude of its item encoding. However, the ωPEV does not make any assumption about the consistency of neural responses over time nor does it take into account a neuron changing its selectivity[23,27,52]. While we show that on the population level, intrinsic timescale affects the stability of information across time, we have no single- cell measure quantifying temporal coding stability for an individual cell. Hence, we chose to evaluate the effective temporal coding dimensionality ($N_{eff}$, where N stands for the number of principal components) of each cell via principal component analysis (PCA; see Methods). In neural population analyses, PCA is classically used for dimensionality reduction; i.e., each neuron counts as a dimension and each trial or time point counts as a point in neuronal state space. Here, we apply the same method to a single

neuron with adjacent time bins counting as dimensions and mean condition firing rates counting as points in "time state space". Specifically, a high $N_{eff}$ indicates a high temporal dimensionality which can be caused by either nonspecific activity, unstable firing, or switching selectivity.

We estimated the $N_{eff}$ in a stepping 500-ms time window composed of 10 independent 50-ms bins (step size of 50 ms). This particular relation between time window and bin size was chosen since it allowed us to evaluate single epochs of the task while capturing most of the temporal coding dimensionality within a given task epoch, and across brain regions (Supplementary Figure 7). As expected, $N_{eff}$ was the highest during the fixation period, when condition-specific firing was absent (Fig. 5a). Eliminating condition-specific firing by shuffling condition labels yielded $N_{eff}$ similar to those of the fixation period. In lPFC and FEF, cells with longer $\tau$ values displayed a lower temporal

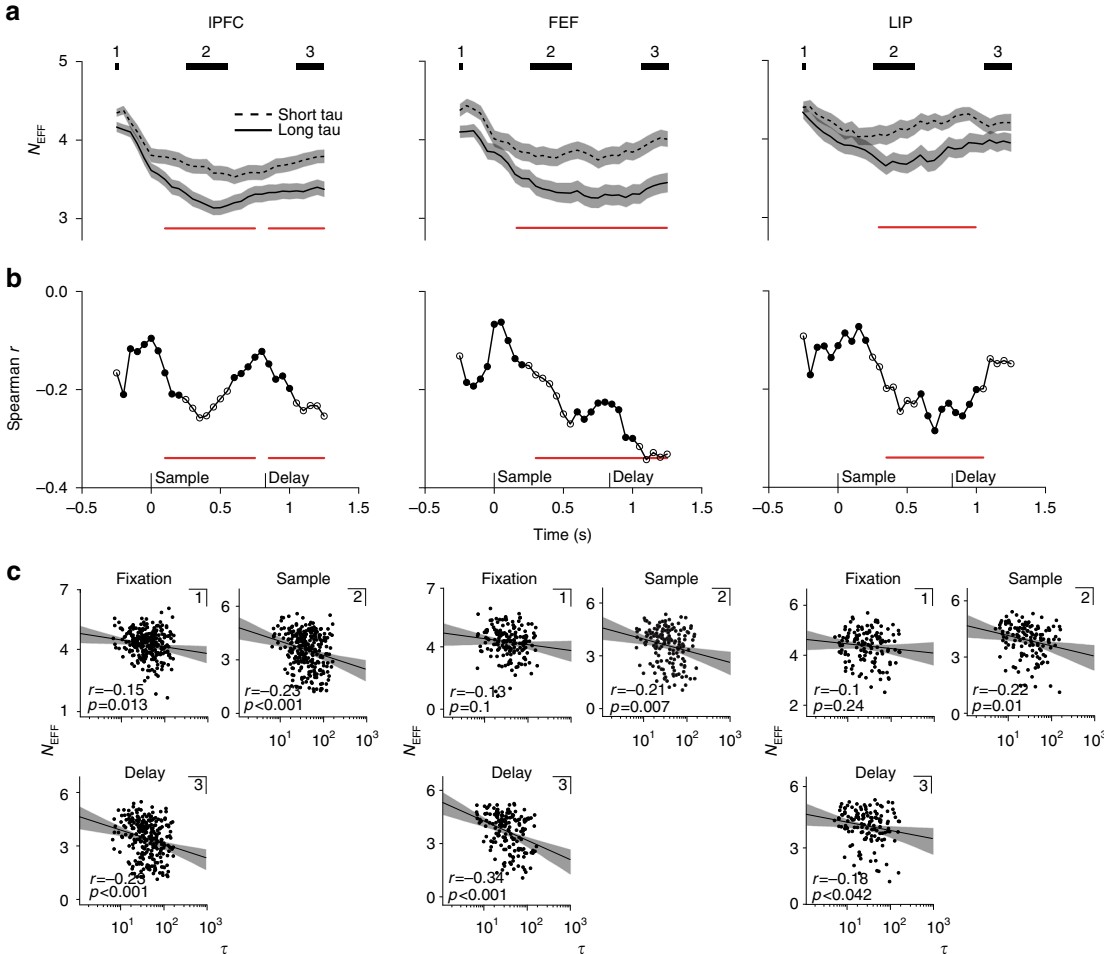

**Fig. 5** Effective temporal dimensionality and intrinsic timescales. **a** Effective temporal dimensionality ($N_{eff}$) estimated at a single-cell level ($N_{eff}$) was estimated from a sliding 50-ms×10 (500 ms) time window shifted by 50 ms on each step. Dashed black line: Average $N_{eff}$ for short $\tau$ cells. Solid black line: Average $N_{eff}$ for long $\tau$ cells. Red horizontal line: significant difference between the two split halves (cluster-based permutation test, $p < 0.01$). Shaded regions mark s.e.m. Black bars mark $N_{eff}$ estimated within independent trial epochs, i.e., where the sliding window did not overlap with distinct epochs (1: fixation; 2: sample; 3: delay). **b** Spearman rank correlation between $N_{eff}$ and log-transformed $\tau$ values. Each dot marks the correlation coefficient as a function of time, i.e., the center of the time window from which $N_{eff}$ was estimated. Red line: significant Spearman rank correlation coefficient (cluster-based permutation test, $p < 0.01$). Hollow dots mark independent trial epochs (1: fixation; 2: sample; 3: delay). **c** Scatter plots for $N_{eff}$ estimated from independent trial epochs (the number in the left corner refers to the number on the black bar) and $\tau$ values. Each dot represents a neuron: lPFC ($n = 264$); FEF ($n = 167$); LIP ($n = 134$)

dimensionality during sample and delay periods (cluster-based permutation test, $p < 0.001$) (Fig. 5a). In LIP, the difference between $N_{eff}$ of the two split halves was significant only around the sample period. The rank correlation time courses in Fig. 5b confirm this observation. In FEF, the correlation between $N_{eff}$ and $\tau$ steadily increases over time, peaking during the delay period (cluster-based permutation test, $p < 0.01$). In lPFC, a strong correlation is evident already early during the sample period ($p < 0.01$), which dips between the two task periods, suggesting a $\tau$-independent state transition, in order to reemerge during the delay period ($p < 0.001$). In LIP, a dependence of $N_{eff}$ on $\tau$ is clearly only present during the sample period ($p < 0.01$). Scatter plots of all neurons/areas in Fig. 5c show the same correlation but for the average $N_{eff}$ estimated from independent time periods within task epochs. Since $N_{eff}$ gives an estimate of the temporal variability of neuronal firing, we should expect to observe a higher $N_{eff}$ for shorter $\tau$ values during the fixation period. In fact, we found such a relationship in lPFC (Fig. 5c; Spearman rank correlation, $r = -0.15$, $p = 0.013$) but failed to observe a significant effect in FEF ($r = -0.13$, $p = 0.1$) and LIP ($r = -0.1$, $p = 0.23$).

**Robust temporal dynamics support discrimination**. Next, we hypothesized that if a short $\tau$ cell genuinely encodes item information, albeit in a temporally dynamic way, the temporal activity pattern should carry information not evident in a purely stable cell. In brief, per neuron, we split trials into two independent halves and quantified the correlation between the temporal pattern of item-specific activity estimated over 200 ms between the two split halves (see Methods). If the temporal activity trace is corrupted by noise, correlation between the split halves (i.e., the discriminability score) should be small. Similarly, a condition-specific but temporally stable trace would also translate into a comparatively small discrimination score. However, a dynamic temporal activity pattern should result in a relatively high discrimination score.

Indeed, item information was discriminable in the temporal coding dynamics inherent in both short and long $\tau$ cells (cluster-based permutation test, $p < 0.001$) (Fig. 6, for lPFC and Supplementary Figure 8, for FEF and LIP). In lPFC, temporal discriminability time courses were strikingly similar during both the sample and test periods. During the delay period, temporal discriminability initially rose but then stayed significant only for

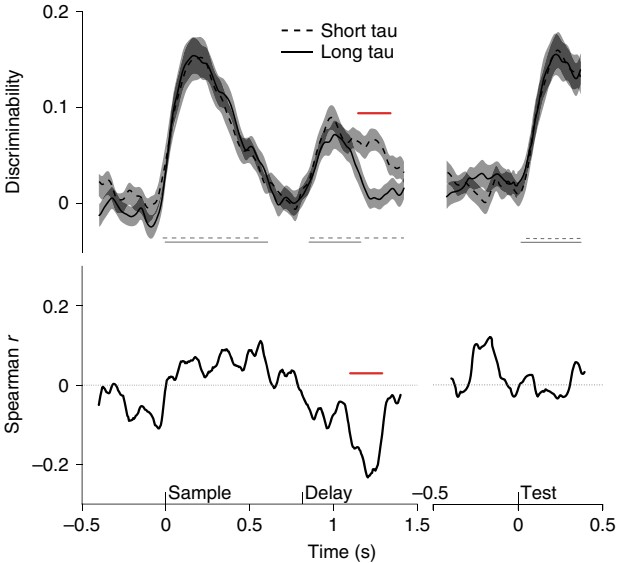

**Fig. 6** Temporal discriminability in lPFC. The upper panel shows the temporal discriminability score (see Methods) averaged over neurons. The black solid line indicates the average over long τ cells and the black dashed line indicates the average over short τ cells. The shaded areas indicate s.e.m. Thin gray lines indicate significant discriminability (cluster-based permutation test, $p < 0.001$), for respective τ splits. The red solid line depicts a significant difference between the two splits of the data (cluster-based permutation test, $p < 0.05$). The lower panel shows the Spearman rank correlation between τ and discriminability score for each time point. The solid red line denotes a significant correlation (cluster-based permutation test, $p < 0.05$)

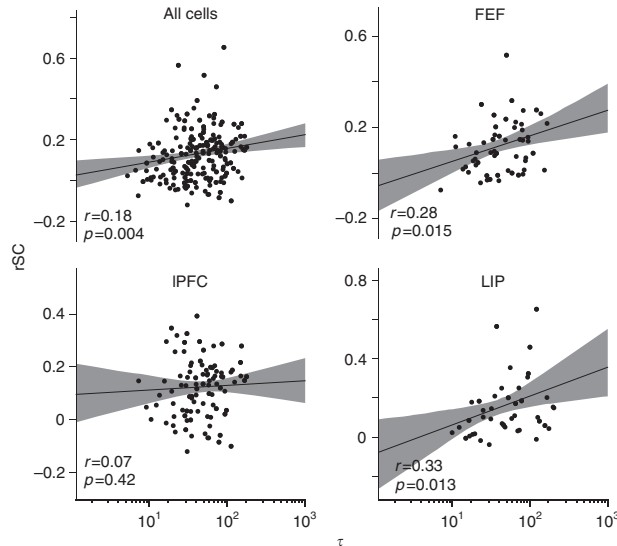

**Fig. 7** Spike count correlations and intrinsic timescales. Scatter plots for rSC on the y-axis versus the τ values on the x-axis. rSC is estimated during the fixation period and between cells on the same contact. Each dot depicts one cell. The r value within panels depicts the Spearman rank correlation coefficient and the associated p-value, for the two variables. All cells: $n = 255$; lPFC: $n = 126$; FEF: $n = 74$; LIP: $n = 55$. Black solid lines show linear regression lines. Shaded region depicts 95% confidence interval

short τ cells ($p < 0.001$), which showed significantly higher decoding during part of the delay ($p = 0.017$). In FEF and LIP, we did not observe any significant difference between the two subpopulations separated by τ (Supplementary Figure 8). Interestingly, in all three brain regions, our method seems to capture transitions between epochs as decoding falls back to baseline as task epochs unfold.

Correlating τ and temporal discriminability over time accentuates the observation made from the median splits. In lPFC, the rank correlation time course shows a significant dip during the delay period (cluster-based permutation test, $p < 0.001$) emphasizing a stronger temporal discriminability for short τ cells (Fig. 6). Interestingly, a similar dip is evident in the FEF correlation time course at a later time point in the delay ($p < 0.01$) which was only hinted at in the median-split plots (Supplementary Figure 8a).

Note, it is unlikely for any cell to encode information in a perfectly stable way and strong item encoding is likely to drive much of the observed correlation. However, keeping in mind the weaker item encoding for short τ cells as observed by the ωPEV, an equal or even stronger temporal discriminability score for the same cells indicates robust but dynamic coding within this subpopulation.

**Functional connectivity correlates with intrinsic timescale.** Next, we tested possible sources leading to our observed τ values. A recent modeling study proposed that intrinsic timescales could, in part, arise through local connection densities[42]. This observation raises the possibility that neurons associated with a long τ value might be part of a local network hub with a high concentration of incoming and outgoing connections[34,39]. To test this hypothesis, we probed functional connectivity using spike

count correlations (rSC), also referred to as noise correlations[53] (see Methods). Spike count correlations were estimated from the fixation period since we wanted to determine whether τ depended on functional connectivity at rest, i.e., before a stimulus synchronized network activity. We found a significant correlation between τ values and rSC when pooling over all brain regions (Spearman rank correlation; $r = 0.18$, $p = 0.004$) (Fig. 7 upper left panel). Spike count correlation can depend on basic firing rate[54]. However, we found no evidence for a significant correlation between rSC or τ and fixation activity, respectively (Supplementary Figure 9; $r = -0.05$, $p = 0.37$; $r = -0.05$, $p = 0.38$). Furthermore, the relationship between rSC and τ was also present when controlling for baseline firing using a multiple linear regression model ($\beta = 0.03$; $p = 0.007$). Together, this implies that differences in fixation activity were not responsible for the positive relationship between rSC and τ. Separating out individual brain regions revealed a more diverse picture: In lPFC, rSC and τ were not correlated ($r = 0.04$, $p = 0.63$, Fig. 7 lower left panel), while both FEF and LIP showed a positive correlation between rSC and τ (FEF: $r = 0.28$, $p = 0.015$; LIP: $r = 0.33$, $p = 0.013$), which held true when controlling for baseline firing using the multiple regression model (FEF: $\beta = 0.05$, $p = 0.02$; LIP: $\beta = 0.04$, $p = 0.02$).

## Discussion
Heterogeneity in temporal tuning observed during WM is related to baseline firing dynamics of individual neurons. Specifically, we found that cells with a longer intrinsic timescale carried more information throughout the task epochs, and particularly during the delay period in prefrontal areas. Important to WM function, mnemonic coding of long-timescale cells was more stable across time in lPFC and FEF. In contrast, short-timescale cells responded more rapidly during encoding in lPFC, and carried rich temporal information. Overall, we identify a functionally relevant heterogeneity in intrinsic timescales of neurons, which enable them to perform complementary computations: rapid encoding or longer-term storage of information.

The core finding of this study is that the baseline temporal stability of individual neurons in lPFC predicts how robustly they code WM-related information, especially during the mnemonic delay period. Early studies of WM focused on persistent activity in lPFC representing specific items in memory[4]; however, more recent studies also reveal more complex dynamics during encoding and maintenance[24–27]. In particular, it has been shown that single cells become active at different time points during the trial, encode information only transiently, possibly multiple times, and/or switch their selectivity over time[22,27,55]. Importantly, we now find that the metric of intrinsic timescales as estimated at rest (i.e., pre-trial fixation period) predicts the coding dynamics during WM.

Recent evidence shows that despite transient dynamics during sample and early delay periods, information in the later delay period is maintained in a relatively stable population code[27]. In addition, a study by Murray et al.[33] showed that during WM, stable and dynamic population codes coexist. Conceptually, in those studies, dynamic and stable mnemonic representations refer to orthogonal subspaces in high-dimensional neural state space[33]. Here, we extend those observations by proposing that they are linked to intrinsic properties of neuronal subpopulations: long-timescale cells more strongly contribute to the stable subspace, whereas short-timescale cells contribute to the dynamic subspace. Our data also show that this is mostly the case during the delay period in prefrontal areas (Fig. 2 and Fig. 4). This contrasts with LIP, where encoding did not significantly differentiate between timescale subpopulations during the delay, where mnemonic coding was absent. Rather, larger-timescale cells in LIP stably encoded more information during the sample period (Fig. 4). A previous study recording from monkey LIP during a simple delayed-response task, showed that baseline firing stability modulated stimulus-specific activity during the delay period[50]. In that study, baseline firing stability was estimated by a relatively coarse metric: The raw correlation value between spike counts within time bins separated by a distinct lag. The tau metric proposed by Murray et al.[37] more elegantly captures a neuron's intrinsic dynamics. First, the metric is not dependent on specific temporal lag used for estimating the temporal autocorrelation, but rather captures its temporal profile over a number of lags. Second, as a second-order property, the tau metric is inherently more robust to differences in absolute neural activity. Tau, therefore, more accurately captures a cell's intrinsic time window of integration than raw autocorrelation at any specific temporal lag. Furthermore, the complexity inherent in our task (color, locations, and load effects) as well as relatively poor encoding within the LIP population might have diluted an otherwise clearer relationship. Using location information as a discriminative feature revealed some cross-temporal generalization extending to the delay period in LIP (Supplementary Figure 4). In FEF, the difference between short- versus long-timescale cells was striking in terms of the magnitude of information encoding during sample and delay period (Figs. 2 and 4). Different from lPFC, the discriminative pattern within the long-timescale subpopulation cross-generalized more strongly from the sample, all the way through the delay period. Note, in our task, strong, temporally stable signals as observed in FEF are unlikely to reflect oculomotor preparation since monkeys cannot prepare a response in advance. Importantly, by analyzing local field potentials (LFP) on a single-trial basis, Lundqvist et al.[24] showed that WM was manifested by the rate of discrete bursts in gamma activity[24]. They argue that observed persistent activity and stable population coding arises through trial averaging of discrete idiosyncratic activity bursts. Although this result might seem contradictory to the temporal stability observed in the long tau cells, it is important to point out that the two results focus on very different

neural signatures (spiking and LFP), at very different temporal scales (100 s vs 1000 s of ms). It would be interesting to see future work that explores the relationship between intrinsic neuronal timescales and LFP oscillations, and their relation to WM. For now, dynamic bursts in both the LFP gamma signal and stable spiking are probable mechanisms underlying WM.

Results from an information-theoretic approach suggest that WM information can be stored more faithfully if the brain first encodes information appropriately before passing it to persistent activity networks[56]. Within the same study, the authors showed that human performance is better fit by such a two-step model than by a model mimicking direct storage in persistent channels[56]. Here we show that the measure of an intrinsic timescale can help explain the differences in temporal dynamics and therefore distinct computational roles within a complex WM task. In lPFC, we saw that stable coding within the long-timescale subpopulation generalizes only weakly across the different epochs of the task (i.e., sample to delay period) (Fig. 4), suggesting that in lPFC, the neural code transitions between different epochs, ultimately to be stored in a more stable format during the delay epoch through cells with longer intrinsic timescales. This seems to be in accordance with previous observations in PFC[26,27] and with the two-step model described by Koyluoglu et al.[56], while emphasizing distinct networks for both perception and storage which are potentially co-present during the sample period[52,57–59]. A similar conclusion had been reached by Murray et al.[33], who suggested orthogonal mnemonic and perceptual representations during the sample period with the latter decaying during the delay period. Contributing to the evidence of distinct roles for cells with different timescales, we found that more dynamic cells, as defined by the intrinsic timescale, coded for item information relatively early during the sample period. This onset cascade early in the sample period implies a more specific role for short-timescale cells since they more rapidly detected the presented stimuli. This result might further play into a two-step model (i.e., rapid perceptual encoding and transfer to a robust stable code) of WM encoding leveraging the presence of a heterogeneous pool of available intrinsic timescales. It is important to note that while discussing different neurons being recruited for certain tasks, those neurons, i.e., their respective timescales lie in a continuum rather than belonging to categorically distinct subpopulations. It therefore seems likely that contributions of individual neurons are weighted according to evolving task demands[2]. Along those lines, functional subpopulations are best thought of as distinct dimensions of the neuronal population rather than specific classes of cell types with distinct roles. In our data, this notion is specifically apparent during the gradual evolvement in item encoding in lPFC (Fig. 2), specifically during the onset cascade, where cells became gradually more engaged as a function of tau (Fig. 3).

High temporal autocorrelation implies temporal stability, which seems well suited for WM. On the other hand, temporal autocorrelation also translates to a lower temporal dimensionality, which necessarily limits the informational capacity of coding over time. A recent, influential study found that high neural dimensionality is crucial for complex behavior[1]. Specifically, the dimensionality of the neural representation is expanded through neurons exhibiting mixed selectivity in response to task factors, maximizing the possibility space for linear classification, i.e., readout of the population activity[1]. This view can be extended to the temporal domain, where dynamic changes in neural activity over time, i.e., switching selectivity are reflected in a high temporal dimensionality. Indeed, the information potential of a highly dynamic network presentation is directly proportional to the statistical independence between time points[60]. Here, we could show that a faster decay of temporal autocorrelation at rest directly translates to a higher coding dimensionality during the

task (Fig. 5). While our measure of temporal coding dimensionality ($N_{eff}$) captures selectivity to task parameters, it also reflects the temporal stability or instability of the tuning profile, extending the ωPEV measure substantially. This is evident in the clear epoch transitions in lPFC: long-timescale cells maintain information in a lower-dimensional temporal state only within each task epoch, not in transitions between epochs. Conversely, short-timescale cells exhibit a higher temporal coding dimensionality even before the start of the delay period. In FEF, the relationship between intrinsic timescale and temporal coding dimensionality steadily increases throughout the task, peaking during the delay period, an observation which adds on the observed ωPEV time courses. Complementing those observations, we found that robust but temporally dynamic coding can discriminate information which would otherwise be lost in a purely stable encoding (Fig. 6). Additional information inherent in temporal dynamics was particularly evident in short-timescale cells in lPFC during the delay. Here, short-timescale cells showed weaker item encoding as quantified by the ωPEV. Nevertheless, their inherently high-dimensional coding properties could be used to decode memory-related information.

Temporal dynamics are especially important within the context of cognitive flexibility. According to the principle of adaptive coding, PFC represents information in a dynamic and therefore flexible manner, where neural resources can be recruited "on the fly", according to behavioral demands[61]. A recent study investigating neural correlates of cognitive flexibility in monkeys found that neurons effectively switching their selectivity over time were most active when cognitive control was demanded[62]. Crucially, most of the variance at the population level could be attributed to time or the interaction between time and trial type, showing that different cue–probe combinations are encoded differently in the distinct trial phases[62]. Furthermore, a recent study in monkeys suggested that the interplay between transient and stable coding guides the evolution of category-based decisions during WM[34]. The advantage of a temporally stable representation during WM seems straightforward, also because such a low-dimensional subspace could be read out over time by a set of fixed weights[33]. On the other hand, such a low-dimensional representation would limit the information capacity for coding transient events. An intrinsic reservoir of continuous neuronal timescales therefore seems particularly well suited for parallelizing either computation, as implied within the present study. In fact, depending on the task at hand, different regions might adjust the weightings of cells with different tau values. As an example: Although the tau distributions were roughly similar between LIP and lPFC (Fig. 1), even the higher tau cells in LIP showed only minimal coding during the delay period (Figs. 2, 3). This suggests that intrinsic timescale might be necessary but not sufficient for involvement in WM. It is possible that LIP might be more involved in other types of maintenance (e.g., a delayed saccade[50]), or integration of perceptual information. Therefore, although tau generally scales with position along the cortical hierarchy[37], there might be some further specificity within equivalent levels of the hierarchy according to the type of memoranda or task. More generally, both the engagement of brain regions along the hierarchy according to diverse temporal integration demands[39,41], and/or their weighting of cells according to tau seem to be at play[47–49] when processing complex cognitive tasks. It will be important to further investigate the recruitment of neurons according to their intrinsic timescales when mnemonic information has to be integrated with flexible task demands.

Our initial quantification of baseline autocorrelation (Fig. 1) adds to the accumulating evidence for a hierarchy of intrinsic timescales in the monkey[36–38,42,49], mouse[39], and human cortex[40,41,63]. Prefrontal areas lPFC and FEF showed longer intrinsic

timescales than LIP, which coincides with their position in the cortical hierarchy[64]. In the temporal domain, longer processing timescales directly relate to the integration or maintenance of information over time[39], which is useful for higher cognitive functions such as decision-making or WM[39,46]. Indeed, prefrontal areas and specifically lPFC, which showed the longest average intrinsic timescale, has served as the major focus for WM research over the past few decades[7] and is generally considered crucial for WM function[65,66].

It is tempting to argue that neurons exhibiting longer intrinsic timescales form part of a continuous attractor network. Classical attractor models of WM persistent activity[17–20] imply functional clustering, i.e., increased connectivity between clusters of similarly tuned neurons, which has been observed and been linked to persistent activity in some tasks[67]. Local connection density per se might be a good predictor of intrinsic timescales and in turn persistent activity[7,39,42]. In a recent study, Chaudhuri and colleagues constructed a large-scale brain model aimed at investigating possible mechanisms leading to a hierarchy of intrinsic timescales similar to the one observed in refs. [37,42]. They found that both long-range connections ("inter-areal loops"), as well as local connection densities determined the relative size of intrinsic timescales across brain areas[37,42]. In line with this, a recent study in mice found that areas with higher functional coupling showed larger processing timescales on the level of the population[39]. Together, these observations raise the possibility that an individual intrinsic timescale might be determined by the concentration of incoming and outgoing connections a given neuron receives, i.e., whether it forms part of a local "network hub"[34]. Our results for FEF and LIP indeed suggest that neurons with longer intrinsic timescales might pertain to such a network hub (Fig. 7). In lPFC, we failed to observe a relationship between a neuron's functional connectivity and intrinsic timescale, possibly indicating more flexible connectivity regimes[29,68]. It is important to note that spike count correlations do not necessarily indicate anatomical connectivity and their source does not have to be local[53]. Finally, there exists a multitude of cell-intrinsic or network mechanisms underlying persistent activity accounts[10]. This plurality naturally applies to possible sources for the heterogeneity of observed intrinsic timescales.

Through analyzing simultaneous electrophysiological recordings from three brain regions: lPFC, FEF, and LIP, we found that the temporal tuning heterogeneity previously observed during WM is predicted by baseline firing stability of individual neurons. We showed that intrinsic timescales effectively shape the temporal dimensionality of the encoded information. In the prefrontal regions lPFC and FEF, neurons with longer intrinsic timescales encoded task information more strongly while storing it in a comparably stable, low-dimensional format throughout the delay. In contrast, cells exhibiting short intrinsic timescales encoded relevant features in a more dynamic fashion, maintaining a high temporal coding dimensionality throughout the task, which in lPFC, could be robustly decoded even during the delay period. In addition, in lPFC, the presented sample stimulus was more rapidly encoded by short-timescale cells. Last, we presented some evidence for distinct mechanisms contributing to the emergence of intrinsic timescale heterogeneity. In summary, we suggest that WM constitutes the dynamic recruitment of neurons with different intrinsic timescales, possibly shaped by local connectivity regimes, to optimize functionally coexistent computations such as the encoding and storage of information.

## Methods

**Experimental paradigm and recordings**. Physiological recordings have previously been reported in detail[51]. In brief, two adult rhesus monkeys (*Macaca mulatta*) were trained to perform a delayed change localization task (Fig. 1a). After a short

fixation period (500 ms), an array of colored squares was presented for 800 ms (the sample period), followed by an 800–1000-ms memory delay (delay period). Finally, a test array was presented to the animal. The test array was identical to the sample array except that one randomly chosen item changed color. To receive reward, monkeys had to make a saccade to the changed item. On each trial, the total number of items on the screen varied between two and five. There were six possible item locations and two possible color values per location (changing each session). Over several sessions, simultaneous recordings were taken from single neurons in the lateral prefrontal cortex (lPFC; 584 cells in total); the frontal eye fields (FEF; 325 cells); and the lateral intraparietal cortex (LIP; 284 cells). All procedures followed the guidelines of the Massachusetts Institute of Technology Committee on Animal Care and the National Institutes of Health.

**Preprocessing**. We initially excluded cells with fewer than 100 spikes per session and less than 50 trials from all analysis, hence reducing the total cell counts per area to 583 cells for lPFC, 323 cells for FEF, and 281 cells for LIP. There were at least 20 trials per relevant stimulus condition. Unless otherwise noted, binary spike trains were convolved with a Gaussian kernel (s.d. = 20 ms) to produce firing rates in sp/s. For the cross-temporal discriminability analysis, firing rates were downsampled from 1000 to 100 Hz, allowing for more efficient computing, while maintaining sufficiently high temporal resolution. All data analysis was implemented in Python using a custom-written code.

**Statistical testing**. Throughout this study, we used a nonparametric cluster-based permutation test[69]. In brief, the method compares some observed test statistic with a constructed null distribution while controlling for multiple comparisons across time. Null distributions were constructed in three ways: (1) By randomly permuting condition labels within cells 1000 times (ωPEV and across time discriminability analysis); (2) By randomly permuting intrinsic timescales across cells 1000 times for comparing median-split intrinsic timescales (i.e., t-test or raw differences) and correlation coefficients. (3) By randomly computing correct and error labels between trials for the correct versus error trial analysis. In general, the relevant test statistic or raw difference was computed for the observed data as well as for each of the 1000 permutations. This was done for each individual time point or a pair of time points (across time discrimination analysis). For (1) and (3), points were classified by comparing the observed effect to the 95th percentile of the null distribution. Contiguous points exceeding the 95th percentile were deemed candidate clusters. For (2), cluster candidates were directly derived from the primary test statistic (contiguous analytical p-values < 0.05 from t-test or correlation analysis). To correct for multiple comparisons, we compared the maximum summed cluster test statistic of the observed data to a new null distribution of maximum summed clusters derived from our permutations. If the size of the observed cluster exceeded the 95th percentile of the new null distribution it was deemed significant.

**Estimation of intrinsic timescales**. Intrinsic timescales (τ) for single cells were estimated from the 500-ms fixation period[37]. To calculate the temporal autocorrelation, spikes were counted in ten 50-ms successive, independent time bins. This procedure created a trial by time-bin matrix. The temporal autocorrelation is the Pearson correlation across trials, between spike counts from each specific time bin; i.e., we computed the correlation of each column of the matrix with every other column. We fitted an exponential decay function to the resulting autocorrelation time course using nonlinear least squares, as implemented by Levenberg–Marquardt algorithm within ScPy's optimize_curve_fit function[37] (Fig. 1b; Eq. 1).

$$R(k\Delta) = A\left[\exp\left(-\frac{k\Delta}{\tau}\right) + B\right] \quad (1)$$

where τ is the intrinsic timescale, A is the amplitude, and B is the offset parameter. The kΔ parameter refers to the relative time lag between time points (50–450 ms). Similar to previous studies[37,49], a fraction of cells showed relatively low autocorrelation values at short time lags, possibly due to the refractory period or negative adaptation. To accommodate for that feature, fitting started at the first reduction in autocorrelation.

Cells were not assigned an intrinsic timescale if they had a fixation period firing rate lower than 1 spike/s or no spikes within any of the 50-ms time bins across all trials, leaving us with the following cell counts per region: lPFC (431/583); FEF (241/323); LIP (231/281). We further excluded cells whose autocorrelation was not well fit by an exponential function, as determined by the following criteria: (1) A first reduction in autocorrelation later than a time lag of 150 ms (lPFC: 7/431; FEF: 2/241; LIP: 15/231). (2) Cells for which the exponential fit was quasi-linear within the measured interval, leading to an overestimation of τ > 500 ms (lPFC: 90/431; FEF: 40/241; LIP: 36/231). (3) Cells that were clearly not fit well by an exponential function, as determined by blinded visual inspection (lPFC: 69/431; FEF: 31/241; LIP: 44/231). In total, this left us with 265 cells from lPFC, 168 cells from FEF, and 136 cells from LIP that were assigned a τ value and hence, were available for further analysis.

**Information content of single cells**. We based our estimation of single-cell information content on the percentage of variance-explained (ωPEV) statistic[51]. Specifically, we were interested in single-cell selectivity to location and/or color information. First, we created a dummy-coded design matrix where each column represented a stimulus condition, i.e., location one (1/6) and color one (1/2). For each trial, a one denoted that that specific condition was met whereas a zero denoted the opposite. We then performed a multiple linear regression with the above-defined conditions as independent factors and the firing rate as a dependent variable. Finally, we calculated ωPEV to quantify the modulation of individual firing rates accounted for by the combination of color and location of the presented stimulus items.

$$\omega^2 = \frac{\text{SS}_{\text{Between groups}} - \text{df} * \text{MSE}}{\text{SS}_{\text{Total}} + \text{MSE}} \quad (2)$$

where $\text{SS}_{\text{Total}}$ denotes the total sum of squares, i.e., total variance, $\text{SS}_{\text{Between Groups}}$ denotes the variance between groups, df are the degrees of freedom, and MSE is the mean-squared error of the model. Statistical significance of single-cell ωPEVs was validated using cluster-based permutation testing (see Statistical testing) with randomly shuffled stimulus conditions.

**Cross-temporal discriminability analysis**. Cross-temporal discriminability of stimulus information on the level of the neural population was assessed using the analysis described in the studies by Stokes et al.[26] and Spaak et al.[27]. First, we randomly assigned each trial to one of two independent data splits. We then computed the mean firing rate over trials per neuron per independent split and per condition (locations×colors = 6 × 2 = 12) and took the pairwise differences between all conditions (66) for each neuron within each independent split. For each pairwise condition difference, we then calculated the Pearson correlation for each time point across neurons between independent splits and averaged the resulting correlation coefficients using Fisher's z-transformation to obtain a single time-resolved discriminability measure. This procedure is analogous to a decoder being trained on split one and time point one (t₁) and being tested on t₁′ in split two. It is straightforward to extend this basic decoder to cross-temporal decoding by computing the correlation between each time point (tₙ) in split one and all (same and other) time points tₙ₊ᵢ′ in split two. The result is a two-dimensional matrix (time by time) representing stimulus discriminability at all time points (on a diagonal) as well as across time points (off-diagonals). Significance of discriminability was assessed through cluster-based permutation testing with randomly shuffled condition labels.

To decode location information only, we defined conditions with respect to the stimulus location, regardless of color (number of conditions = 6; pairwise differences = 15). To decode color information only, we focused on the difference between color conditions within each location (number of conditions = 12; within-location differences = 6).

To decode target-only information, i.e., location and color information about the stimulus that was going to change its identity after the delay period, we performed the same discriminability analysis with one minor change. In brief, mean firing rates were computed over trials per neuron per independent split per condition, with a condition not referring to any stimulus in location x and color y (as before) but to the target only.

For the correct versus error trial analysis, we applied our target-decoding approach as described above. Correct and erroneous trials refer to trials on which the monkey either succeeded or failed in identifying the correct location of the stimulus that had changed color (i.e., the target). First, we counterbalanced correct and error trials while controlling for the overall load (i.e., the number of total stimuli on display) as well as for ipsilateral load (i.e., the number of stimuli on the same visual hemifield as the target stimulus) since this was shown to most strongly affect the monkeys' behavior as well as the neuronal signals in the present data[51]. We then applied our target-decoding approach to correct and error trials separately a 100 times (for 100 counterbalanced subselections of correct and error trials).

In general, due to the exclusion of trials when counterbalancing between correct and error trials, as well as generally fewer trials per condition for the target stimulus only (we required > 8 trials per condition to achieve reliable decoding), a large proportion of cells was excluded that did not meet the trial number criterion in addition to any aforementioned criteria applied to our data. This left us with 60 cells in lPFC; 50 cells in FEF; and 35 cells in LIP for the correct versus error trial analysis. Few cells, few trials, and generally weaker target decoding resulted in statistically unreliable observations in LIP, which could not be interpreted using the presented approach.

**Estimation of temporal coding dimensionality**. Principal component analysis (PCA) was used to estimate the temporal coding dimensionality of single cells. Commonly, PCA is used to explain the variance of population activity in neural state space in response to task features[70]. Here, we sought to apply the same idea to single neurons over time, thereby quantifying the temporal variability of task-dependent firing. First, we constructed a data matrix **Y**: For each trial, we counted raw spikes falling into independent 50-ms time windows (spanning the 2000-ms trial). We then averaged binned spike counts by task condition (locations×colors = 6 × 2 = 12). Columns in **Y** correspond to 10 adjacent, independent time windows

(spanning 500 ms) (i.e., dimensions), and rows to the 12 task condition averages (i.e., samples). PCA is then performed on **Y** to quantify and arrange the variance along orthogonal axes, i.e., principal components in the "space" spanned by the ten independent time bins. The eigenvalues associated with each principal component give us the means to quantify the effective dimensionality ($N_{eff}$)[71] of our temporal state space.

$$N_{eff} = \frac{(\sum \lambda)^2}{\sum \lambda^2} \qquad (3)$$

where $\lambda$ represents the eigenvalues. The $N_{eff}$ measure penalizes small eigenvalues which could arise due to noise. A high $N_{eff}$ (the maximum equals ten in our example) suggests a high independence across time, i.e., a high temporal coding dimensionality. In other words, a cell with a high temporal dimensionality has either an unstable temporal condition selectivity (i.e., noise) or switches its condition selectivity over time.

**Temporal discriminability analysis**. To decode task information in single cells by leveraging variability over time, we applied a temporal discriminability analysis. For each neuron, we randomly assigned each trial to one of two independent data splits. We then computed the mean firing rate over trials per independent split and per condition (locations×colors = 6 × 2 = 12) and took the pairwise differences between all conditions (66) within each independent split. For each pairwise difference, we computed the Pearson correlation between time points from a 200-ms window in split A of the data with the same time points in split B of the data. This was done for all pairwise condition differences and the resulting correlation coefficients were averaged using Fisher's z-transformation. By sliding the 200-ms window along the trials (1-ms increments) and repeating the analysis, we obtained a temporal discriminability score, which captures the information present in the temporal structure of the signal. Temporal variability that is purely driven by noise will result in a low temporal variability score, as will condition-specific but stable firing within 200 ms. Conversely, dynamic but specific firing will result in a comparably high temporal discriminability score.

**Computing spike count correlations**. Spike count correlations (rSC) (also known as noise correlations) (Cohen and Kohn[53]) were calculated from the 500-ms fixation period. We derived rSC values by comparing single-cell responses recorded from the same electrode on the same session. This criterion limited our analysis to electrodes on which more than one cell was recorded. Furthermore, not all cells recorded by an electrode had the same number of trials, hence, we were limited to trials common to all cells recorded on one electrode. Here, we set the minimum inclusion criterion to 20 trials. Last, we only included cells that had an average firing rate greater than 2 sp/s during the fixation period. Together with our sub-selection of cells that were assigned a τ value, we were left with 126 cells for lPFC; 74 cells for FEF; and 55 cells for LIP.

For each trial, we counted all spikes falling into the 500-ms fixation period window. We then normalized spike rates on each trial for both mean firing and slow drifts in neural excitation by computing a z-score for each neuron's firing rate on each trial using a sliding window of ten trials before and after the current trial[72].

$$z_i(k) = \frac{r_i(k) - \mu_i}{\sigma_i} \qquad (4)$$

where $r$ is the firing rate at trial $k$ and $\mu$ and $\sigma$ are the mean and standard deviation of the $i^{th}$ neuron's firing rate estimated from the 21 trials centered on $k$. The Pearson correlation of a neuron's z-scores with those of any other neuron gives the rSC. Ultimately, our goal was to correlate rSC with τ. Therefore, in addition to its τ value, each cell was assigned the Fisher-transformed average rSC of all its pairings on the same contact. If only two cells were recorded on one electrode, we reassigned both cells the mean of their respective log-transformed τ values. Furthermore, to evaluate whether a possible correlation between τ and rSC was confounded by firing rates[54], we used multiple linear regression including the mean of the log-transformed baseline firing rates making up each averaged rSC value (see Results).

**Data availability**. The data that support the findings of this study are available from the corresponding author upon reasonable request.

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

## Acknowledgements

This research was funded by the Biotechnology and Biological Sciences Research Council (BB/M010732/1) and a James S McDonnell Foundation Scholar Award (#220020405) to M.G.S.; ONR (N00014-14-1-0681) and NIMH (R00MH092715) to T.J.B. and NIMH (R37MH087027); The MIT Picower Institute Innovation Fund and ONR MURI grant (N00014-16-1-2832) to E.K.M.; and was supported by the NIHR Oxford Health Biomedical Research Centre. The Wellcome Centre for Integrative Neuroimaging is supported by core funding from the Wellcome Trust (203139/Z/16/Z).

## Author contributions

E.K.M. and T.J.B. conceived the initial study and collected the data. D.F.W. and M.G.S. conceived the current study. D.F.W., M.G.S., and E.S. analyzed and interpreted the data. D.F.W. and M.G.S. wrote and revised the manuscript.

## Additional information

**Competing interests:** The authors declare that they have no competing interests.

