## [Peer Review File · Nature Communications]

Reviewers' comments:

Reviewer #1 (Remarks to the Author):

Wasmuht and colleagues show that the functional role of prefrontal neurons in working memory, i.e. whether they engage in stable or dynamic coding of memory content, depends on a particular neurophysiological property, namely the intrinsic timescale of baseline firing. In monkeys performing a delayed change detection working memory task, the authors found stronger encoding of working memory content in cells with long intrinsic time scales (long tau) than in cells with short intrinsic time scales (short tau), as seen in univariate encoding and population decoding analyses. This was evident specifically for long tau IPFC cells that showed stronger encoding during the delay epoch and long tau FEF cells that showed stronger encoding during the sample and delay epoch. In IPFC, item information was shown to be encoded earlier in the sample epoch by short tau neurons than by long tau neurons. Across all areas, shorter intrinsic time scale at rest could be related to higher temporal dimensionality, i.e. the neurons' capacity for dynamic coding. This property benefits short tau neurons when decoding information integrated over a certain time window (temporal discriminability analysis): in IPFC short tau neurons, information could be decoded equally well as in long tau neurons during the whole task and even better during the delay period. Finally, functional connectivity at rest (as measured by spike-count correlation) was significantly (positively) correlated with tau in FEF and LIP, but not IPFC.

The paper addresses a timely issue that is very relevant to the field of working memory coding. The manuscript is not only conceptually interesting but also well-written and the data is presented in a clear and understandable fashion. The conclusions that the authors draw fit well into the current literature and conceptual models of working memory. In the discussion section, the authors address important issues and do a great job of integrating their findings into established and novel theoretical frameworks. I only have a few comments:

Major points

First, while the time constant of the autocorrelation function of a cell's firing is mathematically well-defined, the authors could comment more specifically on its biological relevance. How does the tau measure relate to or differ from baseline activity maintenance (Nishida et al., 2014)? What are the strengths of the tau measure compared to the "relatively coarse metric" (line 589) used by Nishida et al.? And does the tau measure represent the cell's time window for integration of inputs?

Second, if the tau measure represents a neuron's integration window, how can the lack of relationship between functional connectivity and intrinsic timescales in IPFC be explained? Is the lack of a correlation in IPFC at odds with the network hub idea? Or is the selected measure of spike-count correlations simply too crude? Is there another measure that the authors could try?

Finally, in order to show that the decoded information is a behaviorally relevant read-out, it would be important to show a correlation between the neuronal data and behavior. This could be implemented by an error trial analysis. Is the decoding performance during sample and delay reduced in error trials? This could be shown, for example, using discriminability analysis (similar to Fig. 4) for correct trials vs. error trials.

Minor points

In order to allow other researchers to reproduce the presented work, a few more details are required that are listed below:

- Fig. 1b: What are the individual data points?

- Supp. Fig. 1, line 253: Was there a new tau median split for the subpopulation or were two subpopulations taken from the original two splits? How many cells are in the two groups compared here? How many cells are significant within each of the tau splits?
- Supp. Fig. 5, line 400: Was the time window selected in a way that shows the effect of higher Neff in short tau cells for all three brain regions?
- Fig. 5a: How were the "independent trial epochs" selected? Sliding windows without overlap to other trial epochs? This could be stated more explicitly.

In order to allow the reader to follow the conclusions, a few things could be stated more explicitly:

- References to the corresponding figures in the discussion would allow the reader to track the conclusions more easily.
- Line 395: "mean condition differences firing rates": Does this mean condition differences of firing rates?

Finally, there are a few formal errors:

- Fig. 1a: Caption should say "right panel shows recording sites"
- Fig. 5a: Caption wrong: solid and dashed red lines are in fact black
- References 3 and 11 are the same
- References 36 and 51: title missing

Conclusion

The paper presents a novel application of the intrinsic timescale measure in the context of stable and dynamic coding of working memory content. This work furthers our understanding of how individual cells engage in the mnemonic coding of information and what the properties are that determine their roles. With the few clarifications requested above and one additional piece of supporting evidence (error trial analysis), I would be happy to see this paper published in Nature Communications.

Reviewer #2 (Remarks to the Author):

The study by Wasmuht and colleagues investigates how single unit activity temporal dynamics and timescales in 3 cortical areas in macaque contribute to phasic or sustained encoding of information during a working memory task.

The authors first defined the timescale of each single unit activity recorded in a fixation period and then used this for cell grouping in long vs. short time constant populations. This discrimination was then used to show that long TC cells contribute more to sustained (delay) activity and the encoding of information during the memory delay. They also showed that short time constant populations contributed earlier to stimulus encoding in the sample period. Globally, various levels of analyses related to decoding and measures on temporal dynamics reveal that a range of intrinsic timescales of units contribute to various aspects of computations during working memory task, and in different manners in different cortical areas. They also confirm that both dynamic and stable coding contribute to information processing during delayed response tasks.

Overall this paper is well written and reports very interesting analyses contributing to the currently expanding group of works that identify the dynamical properties of neural networks producing cognition.

Several aspects need to be clarified however which would make the report closer to data, in particular regarding the continuum of timescales. Other aspects might also need revision.

- It seems that information about task condition is missing or poorly reported. When the authors label conditions to decode location or color from cell activity. The task seems to be that monkeys detect a

particular location and color change only after the delay and this location would be, depending on the condition, one of 2 or 6 highlighted locations. So what location information is used for the decoding that can be detected in neural activity from the very beginning of the sample period? Does it reflect configurations of stimuli, or number? if yes 'location' is not really appropriate. In any case this should be better explained.

- The paper in general and the discussion emphasize an intrinsic distinction between functional populations which perform complementary computations. However, and this is actually shortly mentioned in the 2nd page of discussion, none of the described analyses really reveal a distinction between two populations. The real picture is a gradient of timescales and probably of computations. Indeed in most figures coding properties and dynamics of short and long tau populations differ mostly in intensity not in pattern, at least within an area. This is true also for cross-temporal decoding and temporal dimensionality. Would the authors have a way to cluster cells in manners that actually reveal a complementary between computationally different single unit populations? Or else can they expand on the idea of a continuum and be more explicit about it?

- Another maybe related point concerns the fact that the dissociation between unit populations based on timescales and related contributions to information coding, seems to be mostly apparent in LPFC, not in LIP. Yet the distributions of intrinsic timescales in the two structures while statistically different are very comparable, covering the same range of values. The difference probably comes from the weight on short versus long timescales populations in the two structures. Could the authors discuss this, i.e. on how such balance would generate strong differences in global dynamics?

- Also there are discussion points on the putative sources of timescales at the network level whether local or large scale. Early dynamic coding followed by more stable or mix code might reflect the laminar organization of input, processing and output within the cortex. Some of the authors have published data on the laminar dynamics of encoding during delayed response tasks; is there any evidence that could relate the two: laminar organization and differential timescales, or cell types within layers ?

Reviewer #3 (Remarks to the Author):

In this article, Wasmuht et al. examine how neuronal dynamics predict functional roles in working memory. The study is secondary analysis of the previously published dataset (Buschman et al., 2011). Monkeys learned a difficult working memory task and data were recorded from three cortical areas, dorsolateral prefrontal cortex (IPFC), frontal eye fields (FEF), and lateral intraparietal cortex (LIP). The main message of article is intriguing: the time constant of a neuron's spike-count autocorrelation function at rest ("intrinsic timescale") is predictive of information represented during working memory. This is a good addition to the literature but some of the conclusions of the paper are questionable and some additional analysis must be added.

1. Short-tau neurons don't achieve higher information value in sample period than long-tau neurons, in any of three brain areas, in figure 2. But the authors reach opposite conclusion, when they restrict analysis to neurons with significant PEV, and focusing only on IPFC. Even in this case, the difference between short-tau and long-tau neurons is not significant. The only significant (red) time points in supplementary figure 1 are in the delay period. FEF and LIP results also show that early in trial, long-tau neurons still have higher information than low-tau neurons. Authors rely on correlation between tau and PEV computed in an arbitrary interval (50 ms before to 200 ms after onset of sample period, appearing in figure 3) to reach their conclusion. The way selection criterion was framed (significant PEV at ANY TIME during the trial) handicaps the long-tau neurons in this comparison, since long-tau

neurons are more likely to have significant information in delay period and not at interval to test which group is more important in early encoding. Was this correlation negative for FEF and LIP? These are all problems with this analysis. The conclusion that intrinsic timescale predicts encoding onset, just is not well supported by the data and distracts from main message of the paper.

2. Authors set up in introduction alternative paradigms for the maintenance of working memory, including depending on bursts of activity. However their results show that most informative neurons in working memory are those with high temporal autocorrelation, which implies temporal stability. These neurons also exhibit higher spike-count correlation values, which implies regularity from trial to trial. These findings are contradictory with highly dynamic gamma-bursting representing most of the information in working memory. Authors should explain in discussion.

3. The analysis is careful and takes into account average fixation rate, for comparing high- τ and low- τ neurons, for example when they regress PEV against τ . What was the actual fixation rate? Was there difference in firing rate of fixation between long- τ and short- τ neurons? Also, was there difference in firing rate of delay period? It would be good to add a figure of firing rate, to understand how different groups react during trial.

4. Also do neurons with shorter taus have shorter response latencies?

5. Reference #73 is out of order in the text

Thank you for considering our manuscript. Firstly, we thank all reviewers for the fair appraisal of our paper, insightful comment and useful feedback. The corresponding corrections have improved our submission. Below, we address each point (in red), indicating any corresponding changes to the paper (in blue):

Reviewer #1 (Remarks to the Author):

First, while the time constant of the autocorrelation function of a cell's firing is mathematically well-defined, the authors could comment more specifically on its biological relevance. How does the tau measure relate to or differ from baseline activity maintenance (Nishida et al., 2014)? What are the strengths of the tau measure compared to the "relatively coarse metric" (line 589) used by Nishida et al.? And does the tau measure represent the cell's time window for integration of inputs?

Thank you for raising this important issue. The tau measure of intrinsic timescale was originally developed by Murray et al. (2014) to specifically quantify the duration i.e. decay of the temporal autocorrelation, rather than simply the magnitude of the correlation, as used by Nishida et al. (2014). More specifically, Nishida et al. (2014) computed the auto-correlation between time-bins of a specific spacing and used the resulting raw correlation coefficient as their baseline activity maintenance index (BM). First, this index depends on the chosen spacing between the time-bins for which correlations are computed (they used several), and thus can lead to very different estimations of baseline activity maintenance. Considering this, the temporal gradient of the decay function more elegantly captures the intrinsic time-scale of the activity profile (Murray et al., 2014). Second, raw autocorrelation values depend on firing rate [Murray et al., 2014; e.g., which we also show in our data; FEF ($p=0.04$) and LIP ($p<0.001$)]. Here, we exploit the measure developed by Murray et al. (2014) as a better reflection of the cell's time window for integration, taking into account the behavior of the full autocorrelation (for all possible spacings) estimated over the entire fixation period. Given the importance of this point, we have now added a further clarification in the text:

Lines 607-614: "In that study baseline firing stability was estimated by a relatively coarse metric: The raw correlation value between spike counts within time-bins separated by a distinct lag. The tau metric proposed by Murray et al. (2014) more elegantly captures a neurons intrinsic dynamics. Firstly, the metric is not dependent on specific temporal lag used for estimating the temporal autocorrelation, but rather captures its temporal profile over a number of lags. Secondly, as a second-order property, the tau metric is inherently more robust to differences in absolute neural activity. Tau, therefore, more accurately captures a cell's intrinsic time window of integration than raw autocorrelation at any specific temporal lag.

The last part of the added paragraph emphasizes the biological relevance of tau, which we believe (and hope to show) reflects the cell's time window for integration over inputs.

Second, if the tau measure represents a neuron's integration window, how can the lack of relationship between functional connectivity and intrinsic timescales in IPFC be explained? Is the lack of a correlation in IPFC at odds with the network hub idea? Or is the selected measure of spike-count correlations simply too crude? Is there another measure that the authors could try?

Indeed, we did anticipate a significant relationship in IPFC. Lack of evidence could be due to lack of sensitivity in the spike-count correlations, however it was evidently sufficient to detect significant effects in FEF and LIP. Moreover, we have now tried a number of related methods, which all give similar answers. Specifically, we re-analyzed inter-neuronal correlations using cross-correlation (Bair et al., 2001) and population coupling (Okun et al., 2015). In principle, both measures are more sensitive than noise correlations (i.e. spike count correlations); however, results were qualitatively similar. We are cautious about over-interpreting the apparent null effect, beyond mentioning the possibility that it could reflect a different local architecture across these high-level regions (see lines 750-756).

Finally, in order to show that the decoded information is a behaviorally relevant read-out, it would be important to show a correlation between the neuronal data and behavior. This could be implemented by an error trial analysis. Is the decoding performance during sample and delay reduced in error trials? This could be shown, for example, using discriminability analysis (similar to Fig. 4) for correct trials vs. error trials.

Thank you for this relevant suggestion. We have now implemented this analysis, revealing a relative decrease in item-level decoding for the target stimulus on error trials during the delay period in the IPFC (see supplementary material). As suggested, this confirms the behavioural relevance for the neural activity in IPFC (and our decoding approach). There was a similar trend in FEF, but this finer grained analysis was too noisy to interpret for LIP. We have included all these additional results in the supplemental material for completeness (Supp. Fig. 6).

We also now refer to that new figure in the main text in results (lines 392-403) and methods (lines 806-808 & 881-904).

Minor points

We have implemented all the minor corrections as suggested.

In order to allow other researchers to reproduce the presented work, a few more details are required that are listed below:

- Fig. 1b: What are the individual data points?

Each black dot marks the autocorrelation of each time-bin t and any other time-bin t' at the lag specified on the x-axis. We updated the Fig. 1 caption accordingly

- Supp. Fig. 1, line 253: Was there a new tau median split for the subpopulation or were two subpopulations taken from the original two splits? How many cells are in the two groups compared here? How many cells are significant within each of the tau splits?

We apologize for the inadequate description of Supp. Fig. 1. Yes, there was a new tau median split for the cells with significant object information depicted in Supp. Fig. 1. New cell counts within regions are: PFC: n=203; FEF: n=121; LIP: n=53. In the following, we list the number of cells that were significant within the original tau median splits for each region. PFC (long tau: 113/132; short tau: 90/132), FEF (long tau: 63/84; short tau: 58/84) LIP (long tau: 29/67; short tau: 24/67).

We have updated the caption of Supp. Fig. 2 (former Supp. Fig. 1.) and added the following information in the main text to better incorporate the figure in the text:

Lines 246-251: Furthermore, we assessed the proportion of cells with significant item information within our tau splits. Within each region, a larger proportion of long tau cells showed significant ω PEV at any time point in the trial: PFC (long tau: 113/132; short tau: 90/132), FEF (long tau: 63/84; short tau: 58/84), LIP (long tau: 29/67; short tau: 24/67). When focusing on the significant cells only, while performing a new median split by tau within those, average ω PEV traces resembled the ones computed from the original cell population (Supp. Fig. 2).

- *Supp. Fig. 5, line 400: Was the time window selected in a way that shows the effect of higher Neff in short tau cells for all three brain regions?*

To clarify, we now add the following text:

Lines 423-426: This particular relation between time window and bin size was chosen since it allowed us to evaluate single epochs of the task while capturing most of the temporal coding dimensionality within a given task epoch, and across brain regions (Supp. Fig. 6).

- *Fig. 5a: How were the “independent trial epochs” selected? Sliding windows without overlap to other trial epochs? This could be stated more explicitly.*

Yes, this is now explicitly stated in the caption of Fig. 5.

In order to allow the reader to follow the conclusions, a few things could be stated more explicitly:

- *References to the corresponding figures in the discussion would allow the reader to track the conclusions more easily.*

This is a very nice idea which we now implemented throughout the discussion.

- *Line 395: “mean condition differences firing rates”: Does this mean condition differences of firing rates?*

Apologies, it should have read: mean condition firing rates (fixed)

Finally, there are a few formal errors:

- Fig. 1a: Caption should say “right panel shows recording sites” fixed
- Fig. 5a: Caption wrong: solid and dashed red lines are in fact black fixed
- References 3 and 11 are the same fixed
- References 36 and 51: title missing fixed

Reviewer #2 (Remarks to the Author):

- It seems that information about task condition is missing or poorly reported. When the authors label conditions to decode location or color from cell activity. The task seems to be that monkeys detect a particular location and color change only after the delay and this location would be, depending on the condition, one of 2 or 6 highlighted locations. So what location information is used for the decoding that can be detected in neural activity from the very beginning of the sample period? Does it reflect configurations of stimuli, or number? if yes ‘location’ is not really appropriate. In any case this should be better explained.

We apologize for the confusion. We used the colour and location of the memory items (between 2 and 5 items displayed on each trial) as the task-relevant features (i.e., 12 conditions: six possible locations and two possible colors at any locations which were distinct from colors at other locations). ‘Location’ is defined as the location (not number) of the memory stimuli in the initial array. All memory-relevant information is presented at the beginning of the trial, and so in principle is decodable from the onset of the memory array. In response to reviewer 1, we now show that target decodability varies with performance during the delay period in IPFC, confirming the behavioral relevance of the neural signal.

- The paper in general and the discussion emphasize an intrinsic distinction between functional populations which perform complementary computations. However, and this is actually shortly mentioned in the 2nd page of discussion, none of the described analyses really reveal a distinction between two populations. The real picture is a gradient of timescales and probably of computations. Indeed, in most figures coding properties and dynamics of short and long tau populations differ mostly in intensity not in pattern, at least within an area. This is true also for cross-temporal decoding and temporal dimensionality. Would the authors have a way to cluster cells in manners that actually reveal a complementary between computationally different single unit populations? Or else can they expand on the idea of a continuum and be more explicit about it?

Again, we apologize for any confusion. We do not propose distinct neural populations, but rather a distinct functional dimension to the data. Distinct dimensions can be mediated by a gradation in underlying properties at the single unit level (as per Raposo et al. (2014)). We try to be more explicit in our discussion and throughout the paper. Specifically, we changed:

Lines 117-120 in the introduction now reads: Additionally, LPFC cells with shorter intrinsic timescales signaled item information earlier during memory encoding and showed richer dynamics during the delay period, suggesting specific functions along distinct neural dimensions.

Lines 257-260 in the results concerning the onset latency analysis now read: Analogously, we considered whether short τ cells might therefore be faster at encoding at the time around stimulus onset, since such a division would point towards a differential weighting of cells according to τ for either fast perceptual encoding and longer-term storage of information.

Lines 579-581 in the discussion now reads: Overall, we identify a functionally relevant heterogeneity in intrinsic timescales of neurons, which enable them to perform complementary computations: rapid encoding or longer-term storage of information.

We added **lines 663-665** to the part of the discussion discussing the continuum of timescales: In our data this notion is specifically apparent during the gradual evolution in item encoding in LPFC (Fig. 2), specifically during the onset cascade, where cells became gradually more engaged as a function of tau (Fig. 3).

We further added an additional short paragraph to the discussion addressing the reviewer's comment. Since this paragraph also aims at implementing the reviewer's next comment it is listed below.

- Another maybe related point concerns the fact that the dissociation between unit populations based on timescales and related contributions to information coding, seems to be mostly apparent in LPFC, not in LIP. Yet the distributions of intrinsic timescales in the two structures while statistically different are very comparable, covering the same range of values. The difference probably comes from the weight on short versus long timescales populations in the two structures. Could the authors discuss this, i.e. on how such balance would generate strong differences in global dynamics?

Thank you for raising this point. The LIP results are roughly consistent with a coding preference for long tau cells, however as the reviewer notes, the relationship is not as clear as in LPFC. Moreover, as the reviewer notes, this is unlikely due to differences in the distribution of tau. More likely, we suggest that the difference in the LIP analysis is probably due to the overall difference in the decodability of task-relevant information during the delay period (which was poorer relative to LPFC and FEF). We believe inclusion of LIP provides useful insights (especially given previous evidence from parietal cortex Nishida et al. (2014)), but we are cautious of over-interpreting absolute differences. To address the reviewer's comment specifically, we added a note in the discussion cautioning that the presence of long tau might not be sufficient for a role in WM.

Lines 710-721: In fact, depending on the task at hand different regions might adjust weightings of cells with different tau values. As an example: Although the distribution of tau was roughly similar between LIP and LPFC, high tau cells in LIP showed minimal coding during the delay period. This suggests that intrinsic time-scale might be necessary but not sufficient for involvement in WM. It is possible that LIP might be more involved in other types of

maintenance (e.g., delayed saccade (Nishida et al., 2014)), or integration of perceptual information. Therefore, although tau generally scales with position along the cortical hierarchy (Murray et al., 2014), there might be some further specificity within equivalent levels of the hierarchy according to type of memoranda or task. More generally, both the engagement of brain regions along the hierarchy according to diverse temporal integration demands (Honey et al., 2012, Runyan et al., 2017), and/or their weighting of cells according to tau seem to be at play (Scott et al., 2017, Bernacchia et al., 2014, Cavanagh et al. 2016) when processing complex cognitive tasks.

- Also there are discussion points on the putative sources of timescales at the network level whether local or large scale. Early dynamic coding followed by more stable or mix code might reflect the laminar organization of input, processing and output within the cortex. Some of the authors have published data on the laminar dynamics of encoding during delayed response tasks; is there any evidence that could relate the two: laminar organization and differential timescales, or cell types within layers?

The type of electrodes used in the present study unfortunately do not allow us to distinguish between different layers. It can nonetheless be interesting to speculate on the relationship between laminar dynamics and differential timescales. For example, a recent study (Bastos et al., 2018) found that WM delay activity is mainly evident in gamma-band activity in superficial layers, which in turn is modulated by deep-layer alpha and beta band activity. It is tempting to speculate that the long-timescale cells, which we observed to show the strongest delay activity, are primarily superficial-layer cells involved in gamma bursting. However, due to the limitations of our data and the highly tentative nature of this hypothesis, we believe this goes beyond the scope of the current manuscript, and would be very interesting material for a follow-up study.

Furthermore, different cell types might contribute to the heterogeneity of the observed tau values in our sample. However, we are not classifying different cell types within this study nor are we confident in making any specific predictions within the context of this study. Nevertheless, we do think that relating specific types of cells with tau and WM properties is a crucial next step in future work.

Reviewer #3 (Remarks to the Author):

1. Short-tau neurons don't achieve higher information value in sample period than long-tau neurons, in any of three brain areas, in figure 2. But the authors reach opposite conclusion, when they restrict analysis to neurons with significant PEV, and focusing only on LPFC. Even in this case, the difference between short-tau and long-tau neurons is not significant. The only significant (red) time points in supplementary figure 1 are in the delay period. FEF and LIP results also show that early in trial, long-tau neurons still have higher information than low-tau neurons. Authors rely on correlation between tau and PEV computed in an arbitrary interval (50 ms before to 200 ms after onset of sample period, appearing in figure 3) to reach their conclusion.

We apologize for giving the wrong impression. The reviewer is correct - there was no benefit for short tau cells in the first (single neuron based) PEV analysis, although there was a reliable difference in the (population based) decoding (see Fig. 4a). The purpose of the onset analysis was not to determine whether short tau cells have higher object encoding but whether there is a difference in latencies regardless of the absolute value. We choose the interval -50 ms to 200 ms around onset of sample period to capture the main cascade during the very beginning of the sample period (whilst also making sure that the window was long enough to capture the onset of most cells). Here we further show that our conclusions are independent of the specific choice of interval (Fig. I).

Figure I: Extended coding onset analysis in IPFC. The time course in the lower panel represents the Spearman correlation between coding onset in ms (i.e. time point of first significant PEV, computed over the whole sample period) and tau values in ms; 1ms step-size. The instantaneous correlation is computed from cells whose coding onset fell between 0 (sample onset) and the time point on the x-axis [note: correlation coefficients towards the beginning of the trial are made up of fewer cells, since many cells only became active with increasing trial time]. Correlations were only computed from the time point when >10 cells had shown a significant coding onset (~50ms). The light blue solid line marks the median correlation coefficient obtained from 1000 random permutations (permuting tau values) and the dashed light blue lines mark the 95% confidence interval. Note, also here, correlation coefficients at earlier time points are computed from a lower number of cells. The red solid line marks a significant correlation coefficient ($p < 0.05$; permutation test). Inset scatter plots are the same as in Fig. 3c. Their position in the sample period time course is marked with a red dot. The middle inset depicts the average PEV for cells (median split by tau) that had shown a significant coding onset up to the time point marked by the first red dot (200ms).

The way selection criterion was framed (significant PEV at ANY TIME during the trial) handicaps the long-tau neurons in this comparison, since long-tau neurons are more likely to have significant information in delay period and not at interval to test which group is more important in early encoding.

We apologize for some confusion in our description of the analysis. It is correct in the figure caption but wrong in the text. It should read: “Significant PEV at any time during the sample period since we are only interested in the encoding of the stimulus”. This has been fixed now.

It is empirically true that high tau cells are more likely to be significant during delay. However, this does not handicap the analysis during the sample period, as can be seen from the red line indicating the mean tau value of active cells at any time-point (Fig. 3b). Very many long tau cells are also active during the sample period. It is also important to keep in mind that the long and short tau in this analysis are not always the same cells (correlation values or splits are calculated within each time window (from 0 to point on x-axis)) which emphasizes a continuum of timescales rather than distinct subpopulations as we emphasize more strongly now.

Was this correlation negative for FEF and LIP?

Not significantly (see Supplemental Figure 2 and Fig. II for further clarification)

Figure II: Extended coding onset analysis. This figure depicts the same analysis as in Fig. I, for FEF and LIP. None of the correlations coefficients or differences in PEV were significant.

2. *Authors set up in introduction alternative paradigms for the maintenance of working memory, including depending on bursts of activity. However, their results show that most informative neurons in working memory are those with high temporal autocorrelation, which implies temporal stability. These neurons also exhibit higher spike-count correlation values, which implies regularity from trial to trial. These findings are contradictory with highly dynamic gamma-bursting representing most of the information in working memory. Authors should explain in discussion.*

We are not sure whether we follow the reviewer's argument regarding spike count correlations implying regularity from trial to trial. High spike count correlations rather indicate high covariance of trial-wise responses between different cells (which actually depends on (shared) variance over trials). However, the reviewer raises an important point when referring to gamma bursting during WM. We added a short paragraph in the discussion to address this point.

Lines 624-634: Importantly, by analyzing local field potentials (LFP) on a single trial basis, Lundqvist et al. (2016) showed that WM was related to the rate of discrete bursts in gamma activity (Lundqvist et al., 2016). They argue that observed persistent activity and stable population coding arises through trial averaging of discrete idiosyncratic activity bursts. Although this result might seem contradictory to the temporal stability observed in the long tau cells, it is important to point out that the two results focus on very different neural signatures (spiking and LFP), at very different temporal scales (100s vs 1000s of ms). It would be interesting to see future work explore the relationship between intrinsic neuronal timescales and LFP oscillations, and their relation to WM. For now, dynamic bursts in the LFP gamma signal and stable spiking are both probable mechanisms underlying WM

3. *The analysis is careful and takes into account average fixation rate, for comparing high-tau and low-tau neurons, for example when they regress PEV against tau. What was the actual fixation rate? Was there difference in firing rate of fixation between long-tau and short-tau neurons? Also, was there difference in firing rate of delay period? It would be good to add a figure of firing rate, to understand how different groups react during trial.*

Thank you for raising those important points. We now add a new supplementary figure to the paper (Supp. Fig. 1) depicting firing rates split by tau, per epoch and region as well as scatter plots of tau and firing rates with associated correlation values. In short, we did not find any significant relationship between raw firing rate and tau in any task epoch or region, although overall, long tau cells show a trend for slightly higher firing rates.

We also added a sentence integrating the figures into the results:

Line 186-187: "...nor was there any significant relationship between tau and firing rate within each task epoch (Supp. Fig. 1)".

4. *Also do neurons with shorter taus have shorter response latencies?*

This is an interesting question. Indeed, we would have expected such a correlation between tau and response latencies. Although there was a clear trend following the PEV onset analysis (see Fig. III below), it was not statistically significant.

Figure III: Response latency analysis. We computed response latencies as the first time-point during the sample period when an individual neuron's firing rate (trial averaged) became different from its baseline firing (as determined by a cluster base permutation test; $p < 0.05$). The timecourse in the lower panel represents the Spearman correlation between response latency in ms and tau values in ms; 1ms stepsize. The instantaneous correlation is computed from cells whose response latency fell between 0 (sample onset) and the timepoint on the x-axis (note: correlation coefficients towards the beginning of the trial are made up of fewer cells, since many cells only became active with increasing trial time). Correlations were only computed from the time point when >10 cells had shown a significant coding onset (~ 40 ms). The light blue solid line marks the median correlation coefficient obtained from 1000 random permutations (permuting tau values) and the dashed light blue lines mark the 95% confidence interval. Note, also here, correlation coefficients at earlier time points are computed from a lower number of cells. The red solid line marks a significant correlation coefficient ($p < 0.05$; permutation test). The middle inset depicts the average firing for cells (median split by tau) that had shown a significant response up to the timepoint marked by the first red dot (150ms). The red dot marks the timepoint at which 80% of the maximal number of active cells had become active (similar to the PEV analysis for consistency).

5. Reference #73 is out of order in the text

Thanks, fixed.

REVIEWERS' COMMENTS:

Reviewer #1 (Remarks to the Author):

The authors have addressed all my comments satisfactorily. I am happy to support publication of the manuscript in its present form.

Reviewer #2 (Remarks to the Author):

The revision has improved the paper very significantly. The authors globally answered all questions appropriately.

I would just suggest to add more information on number of cells for which the fit with exponential function did not work. The current analysis, that has been chosen by this and a few recent studies, selects units based on this criteria. Yet the numbers provided here reveal that the criteria eliminated half of the recorded cells. Not taking into account the non spiking or enough sampled units, it remains probably different types of functional units, like oscillating spiking, cells with linear autocorrelation, etc. So giving an idea of what fraction of neurons we miss by selecting data based on an exponential function would be interesting.

Reviewer #3 (Remarks to the Author):

The revised manuscript has addressed my concerns.

We thank all reviewers for the fair appraisal of our paper, insightful comment and useful feedback. Below, we address each point (in red), indicating any corresponding changes to the manuscript (in blue):

Reviewer #1 (Remarks to the Author):

- The authors have addressed all my comments satisfactorily. I am happy to support publication of the manuscript in its present form.

Thank you for reviewing our revised version and supporting its publication.

Reviewer #2 (Remarks to the Author):

- The revision has improved the paper very significantly. The authors globally answered all questions appropriately.

- I would just suggest to add more information on number of cells for which the fit with exponential function did not work. The current analysis, that has been chosen by this and a few recent studies, selects units based on this criteria. Yet the numbers provided here reveal that the criteria eliminated half of the recorded cells. Not taking into account the non spiking or enough sampled units, it remains probably different types of functional units, like oscillating spiking, cells with linear autocorrelation, etc. So giving an idea of what fraction of neurons we miss by selecting data based on an exponential function would be interesting.

Thank you for raising this issue. We now provide a more detailed account of the proportion of cells that were excluded during each step.

Lines 805-806 now reads: We initially excluded cells with fewer than 100 spikes per session and less than 50 trials from all analysis, ...

Lines 847-858 now read: Cells were not assigned an intrinsic timescale if they had: A fixation period firing rate lower than 1 spikes/s or no spikes within any of the 50ms time bins across all trials, leaving us with the following cell counts per region: IPFC (431/583); FEF (241/323); LIP (231/281). We further excluded cells whose autocorrelation was not well fit by an exponential function as determined by the following criteria: (1) A first reduction in autocorrelation later than a time-lag of 150ms (IPFC: 7/431; FEF: 2/241; LIP: 15/231). (2) Cells for which the exponential fit was quasi-linear within the measured interval, leading to an overestimation of $\tau > 500$ ms (IPFC: 90/431; FEF: 40/241; LIP: 36/231). (3) Cells that were clearly not fit well by an exponential function, as determined by blinded visual inspection (IPFC: 69/431; FEF: 31/241; LIP: 44/231). In total, this left us with: 265 cells from IPFC, 168 cells from FEF and 136 cells from LIP that were assigned a τ value and hence, were available for further analysis.

Reviewer #3 (Remarks to the Author):

- The revised manuscript has addressed my concerns

Thank you for reviewing our revised version and supporting its publication.